# Metal Oxide Chemiresistors: A Structural and Functional Comparison between Nanowires and Nanoparticles

**DOI:** 10.3390/s22093351

**Published:** 2022-04-27

**Authors:** Andrea Ponzoni

**Affiliations:** 1National Institute of Optics (INO) Unit of Brescia, National Research Council (CNR), 25123 Brescia, Italy; andrea.ponzoni@ino.cnr.it; Tel.: +39-030-3711440; 2National Institute of Optics (INO) Unit of Lecco, National Research Council (CNR), 23900 Lecco, Italy

**Keywords:** metal oxides, nanowires, nanoparticles, chemiresistors, self-heating, surface functionalization, surface termination

## Abstract

Metal oxide nanowires have become popular materials in gas sensing, and more generally in the field of electronic and optoelectronic devices. This is thanks to their unique structural and morphological features, namely their single-crystalline structure, their nano-sized diameter and their highly anisotropic shape, i.e., a large length-to-diameter aspect ratio. About twenty years have passed since the first publication proposing their suitability for gas sensors, and a rapidly increasing number of papers addressing the understanding and the exploitation of these materials in chemosensing have been published. Considering the remarkable progress achieved so far, the present paper aims at reviewing these results, emphasizing the comparison with state-of-the-art nanoparticle-based materials. The goal is to highlight, wherever possible, how results may be related to the particular features of one or the other morphology, what is effectively unique to nanowires and what can be obtained by both. Transduction, receptor and utility-factor functions, doping, and the addition of inorganic and organic coatings will be discussed on the basis of the structural and morphological features that have stimulated this field of research since its early stage.

## 1. Introduction

Metal oxide nanowires have become a popular class of nanostructures for the development of electronic and optoelectronic devices. Their single-crystalline structure free from extended defects, together with their nm-sized cross section and μm-sized length, make them optimal candidates to merge the potentialities of the nanoscale with efficient electrical transport over much longer distances, with no constrictions from grain boundaries, which typically affect films composed by nanostructures with spherical shapes. In 2001, a fundamental milestone for the technological exploitation of nanowires was reached with the development of a cheap method suitable to synthesize metal oxide nanowires in large amounts and with controlled length, cross-section shape and size [1]. About one year later, these materials were exploited for the first time to develop chemiresistors based on a disordered network of SnO_2_ nanowires [2].

These papers inspired intense research in the field of gas sensors, and more generally, in the areas of electronics and optoelectronics. The field has rapidly grown and a broad range of papers have been published about all the relevant aspects of this technological branch. Dedicated reviews have been and are still being regularly published to track the progress in the synthesis of these nanostructures [3,4,5], their effective integration into functional substrates and devices [6,7,8] and their exploitation in different applicative fields [9,10,11].

This witnesses the potentialities and the appeal of these materials for science and technology in a broad sense, and specifically for chemosensing. About 20 years after the aforementioned pioneering papers, the present work aims at reviewing the state of the art of nanowire-based chemiresistors, emphasizing the comparison with nanoparticles. These are the traditional materials employed in the field but are also the target of recent research. In particular, the manuscript will continue with a section dedicated to the structural and morphological features of nanowires and nanoparticles, from which their potentialities and suitability for technological solutions stem. Synthesis methods will also briefly be reviewed. Gas-sensing models will then be introduced, starting from the knowledge developed with nanoparticles and showing how these concepts have been extended to properly account for shape effects. Finally, some relevant experimental results will be reported to highlight how ideas initially developed for nanoparticles have been further developed to work with nanowires and to discuss those solutions that are unique either to nanoparticles or nanowires.

## 2. Nanowires: Structure, Synthesis and Gas-Sensor Configurations

This paragraph will briefly overview the structural and morphological features of nanowires, highlighting the differences with respect to nanoparticles. Synthesis methods to prepare single-crystalline nanowires and chemiresistor layouts adopted to exploit these nanostructures will also be resumed. Electrical transport across the circuit elements of chemiresistors based on metal oxide (MOX) nanostructures will finally be introduced in this section. Their dependence from the elementary crystallites will be discussed as the basis for gas-sensing models, and results summarized in Section 3 and Section 4, respectively.

### 2.1. Structural and Morphological Features of Nanowires

The high crystalline quality and the well-defined morphology of nanowires has been widely reported in the literature, particularly by means of electron microscopy techniques, emphasizing the novelty of these features with respect to traditional nanoparticles. An overview of the morphological features of nanowires is reported in Figure 1a–c; images of nanoparticles are reported for comparison in Figure 1d,e. The large length-to-diameter aspect ratio, the absence of grain boundaries through the whole length of the wire, and the well-defined surface termination are the main differences with respect to nanoparticles and nanoparticle assemblies. Nanowires used in gas sensing are typically a few μm or tens of μm long; their cross section has a well-defined polygonal shape, typically a rectangle, a square or a hexagon, with a size of the order of a few tens of nm. The axis of the wire corresponds to a crystalline direction; the exposed surfaces correspond to crystalline planes and are often found to feature an almost atomically flat termination. In addition to nanowires, terms such as nanobelts, nanorods are also used to refer to elongated, single-crystalline nanostructures. A nanobelt is typically used for crystallites with a rectangular cross section, such as those shown in Figure 1; nanowire is used in case of a nearly squared cross section; while nanorod is used for single crystals with a smaller length-to-diameter aspect ratio, though much larger than unity. Nanocubes and nanoprisms will also be considered in this review; these are interesting as links between the ideal nanoparticle and nanowire morphologies. Indeed, nanocubes and nanoprisms feature an almost isotropic shape, typical of nanoparticles, together with faceted surfaces, typical of nanowires.

Before to conclude this section, it is worth mentioning that the name nanowire, or porous nanowire, is sometimes used in the literature to indicate elongated structures composed by assemblies of elementary spherical nanoparticles. In the present work, these nanostructures will not be considered in the family of nanowires; this is because they lack a single-crystalline nature, and concerning electrical properties, their electrical transport within each individual assembly is affected by grain boundaries. Owing to these characteristics, they will be regarded as members of another family of materials, namely the hierarchical assemblies, and will be used for comparison with single-crystalline nanowires.

### 2.2. Synthesis of Metal Oxide Nanowires

The first paper about nanowire chemiresistors employed a disordered network of SnO_2_ nanowires synthesized through the vapor–solid (VS) mechanism [2]. The setup was based on a tubular furnace in which a temperature gradient was realized. MOX powders were used as source material and placed in the warmest region of the tube. The vacuum background was of the order of 10 mbar, and the temperature in this region was adjusted to sublimate the powders (around 1350 °C). An Ar flow was used to transport vapors toward the colder region where condensation over substrates occurred [1].

A similar setup has also been widely used for the synthesis through the vapor–liquid–solid (VLS) mechanism. In this case, the substrate is precoated by metallic nanoparticles that catalyze the growth of the nanowire structures. The growth of the oxide nanowire arises from the formation of a eutectic alloy between the metal and the vapor species. As a consequence of the continuous fed of vapors, the alloy reaches a supersaturation condition and the MOX nanowire grows below the alloy nanoparticle [1]. A schematic representation of the VLS mechanism is provided in Figure 2a. The choice of the catalytic nanoparticles, their size and density over the substrate are useful parameters to control the diameter, distribution and density of nanowires [15,16].

Chemical vapor deposition (CVD) techniques have been developed to synthesize nanowires via VS and VLS mechanisms [17,18,19]. With respect to the aforementioned physical methods, CVD techniques allow for the reduction of the synthesis temperature since they do not require the sublimation of the source powders.

To match the requirements of different device layouts, techniques for deposition over selected areas have been further developed. These include lift-off processes employing high-temperature resists [20], the patterning of the catalyst (for VLS growth) [21] or of the substrate roughness (for VS. growth) [22]. In some papers, the electrodes of the gas-sensor device have been used as catalysts to selectively promote the growth of the nanowires directly from the electrodes [18].

An additional effective method for the patterned growth of nanowires is thermal oxidation. In this case, a metallic film is initially deposited over the desired regions and it is further oxidized at high temperature in an oxidative atmosphere to induce the growth of wire-shaped nanostructures [23,24]. It is worth mentioning that the process often starts with the formation of a polycrystalline oxide layer from the metallic film. This polycrystalline layer is typically composed by rounded grains and nanowires grow out from this layer in a second time, as the oxidation process continues. In principle, a residual polycrystalline film may remain beneath the nanowires, resulting in a device whose properties are given by both morphologies. A similar situation may also occur with nanowires grown by VS and VLS mechanisms directly over the sensor substrate. Indeed, if the synthesis is not properly controlled, spurious depositions may occur, and in the worst cases, a continuous polycrystalline layer may deposit under the nanowire mat. In turn, the undesired film may introduce a non-negligible contribution to the final electrical and sensing properties of the device. In these situations, electron microscopy investigation of gently scratched areas and imaging of cross-section samples are useful methods employed to assess whether the electrical and gas-sensing properties of the device are dominated by nanowires or nanoparticles [20,25]. An example is shown in Figure 2b. In this specific case, the nanoparticles underlying the nanowire network were disconnected one another and they were reasonably supposed to not contribute to the electrical and sensing properties of the device [25].

**Figure 2 sensors-22-03351-f002:**
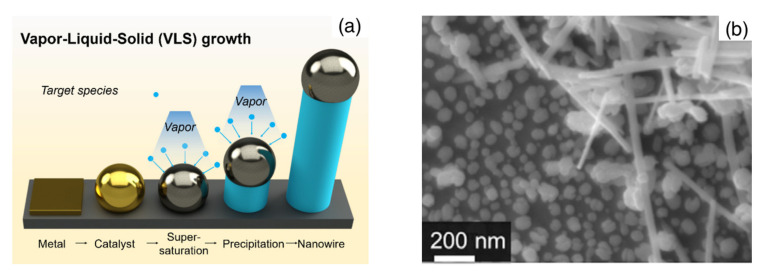
(**a**) Schematic representation of the vapor–liquid–solid (VLS) growth mechanism employed in the preparation of single-crystalline nanowires; (**b**) SEM image showing the residual nanoparticles underlying a network of WO_3_ nanorods synthesized by oxidation of a polycrystalline film. In this case, nanoparticles are disconnected and are not expected to contribute to the electrical and sensing properties of the macroscopic layer. Figure 2a is from [26]; Figure 2b is reprinted from [25], Copyright (2011), with permission from Elsevier.

Wet chemistry techniques, such as, for example, hydrothermal and sol-gel methods, have also been developed to prepare nanowires. These are appealing approaches owing to their reduced cost, low synthesis temperature and the possibility to finely control the nanowire properties, including the addition of dopants, by properly selecting the basic chemicals and tuning the solution composition [27,28]. Gas sensors are typically realized by collecting the nanowires from the synthesis apparatus, dispersing them into a solution or a paste that is further directly applied over the sensor substrate [29,30] or used as ink for screen printing [31]. In some cases, MOX nanoparticles have been used as seed crystals to promote the selective growth of nanowires directly over the desired areas of the sensor substrate [32,33].

### 2.3. Nanowire-Based Chemiresistors: Device Configurations

The simplest way to exploit nanowires in chemiresistor devices is by dispersing a disordered ensemble of these nanostructures over a substrate already provided with electrodes and the heating element. A schematic representation of such a kind of device is shown in Figure 3a together with the main components of its equivalent electrical circuit. Identifying the role of these components in the electrical transport of the device is fundamental to understand its sensing properties, which may be regarded as a gas-induced perturbation to the transport properties of the device. In the equivalent circuit, three main components can be found: nanowires, nanowire–nanowire junctions and the nanowire–electrode contacts, each represented by its equivalent resistor, namely R_nw_, R_j_ and R_c_ [34].

In principle, these three elements are the same as in chemiresistors based on nanoparticles; nonetheless, the different morphological features of spherical and wire-shaped nanostructures introduce differences in the relative weight of these equivalent resistors. Indeed, for nanoparticles, it is widely reported that junctions dominate over crystallites, R_j_ >> R_np_, where R_np_ is the resistance of nanoparticles and R_j_ is more often named the grain-boundary resistance. Similarly, considering the large number of grain boundaries in nanoparticle layers, it is often found that R_j_ >> R_c_. As a result, at least in a first approximation, grain boundaries are usually regarded as the dominant elements in nanoparticle-based devices [35,36].

As far as wire-shaped crystallites are considered, there is a large amount of experimental and computational evidence that electrical transport may be no more junction-dominated and crystallites may play a non-negligible role [37,38,39,40]. As will be discussed in Section 3, this opens the interesting prospective for the exploitation of the sensing properties of the nanowire body, in addition to nanowire–nanowire junctions. Moreover, considering the reduced number of junctions in the network with respect to the nanoparticle networks, the condition R_j_ >> R_c_ may be no more valid. This will be particularly important in the case of non-Ohmic electrode–semiconductor contacts. In this situation, the reversely biased contact may feature a contact resistance, R_cr_, that may be comparable with or even larger than R_j_ and R_nw_ [19]. The role of electrodes in metal oxide chemiresistors has been recently reviewed based on results achieved with thin-film and thick-film technologies [41]. It provides a complete overview of the different conduction regimes that may take place at the metal–semiconductor interface and the related equations, which are also valid in the case of nanowires.

Another important configuration exploited with nanowire materials is the chemiresistor based on a single nanowire, in which the two electrodes are directly connected through the metal oxide nanowire. A schematic representation of this kind of device is shown in Figure 3. This class of devices is very appealing for both fundamental studies and applications, which will be discussed in the next paragraphs.

Before addressing these arguments, it is worth briefly discussing the electrical circuit of the single-nanowire device. The absence of nanowire–nanowire junctions and the high crystalline quality of the nanowire ease the electrical transport through the semiconducting material. This means that the value of R_nw_ may become comparable with, if not lower than, R_cr_ [42]. To decouple R_nw_ from R_cr_, the electrical characterization is often carried out using the four-probe configuration instead of the two-probe one [19,43]. As an example, R_nw_ ≈ 76 MΩ and R_cr_ ≈ 200 MΩ (for an applied voltage of 1 v) were measured at room temperature with a device based on a single SnO_2_ nanowire. The wire had a diameter of 50 nm and was placed orthogonally between two parallel Pt electrodes spaced by about 5 μm [19].

## 3. Gas-Sensing Mechanism

A chemiresistor is a gas sensor based on a sensitive layer that undergoes a variation in its electrical properties upon the interaction with gaseous molecules. The basic concepts underlying the working mechanism of these devices were mainly developed working with thick-film materials and have later been extended to properly account for the nanowire morphology.

The sensing mechanism is usually schematized by means of two functions: the receptor function, which recognizes a chemical substance at the surface of the semiconducting MOX; and the transducer function, which transduces the chemical reactions at the semiconductor surface into the electric output signal [44]. The utility factor, namely the effectiveness of the layer in allowing a proper diffusion of gas through the layer itself, has also been considered in more recent years [45].

These three concepts will be separately introduced in Section 3.1, Section 3.2 and Section 3.3 as the basis of the framework underlying the scientific and technological solutions explored in the literature of nanowire-based chemiresistors.

### 3.1. Receptor Function

Red-ox reactions occurring between the gas molecules and the MOX layer represent the widest-acknowledged phenomena underlying the receptor function of MOX materials.

In these interactions, a key role is played by oxygen ions populating the surface of any metal oxide exposed to ambient air.

At low temperatures (below 150 °C), oxygen is mainly adsorbed molecularly, either in its neutral physisorbed form ($O_{2}$), or in its ionic chemisorbed form ($O_{2}^{-}$). Further increasing the temperature, the population of oxygen species becomes dominated by chemisorbed ions in their atomic forms, namely $O^{-}$ or $O^{2-}$, the latter dominating above 400–450 °C [35].

With the aim to focus on aspects responsible for the modulation of the electrical properties of the oxide material, gas-sensing papers typically adopt the following scheme to describe the chemisorption process [35]:(1)$$\frac{\beta }{2}O_{2,\mathrm{gas}}^{}+\alpha e^{-}↔{\;O}_{\beta ,\mathrm{surf}}^{-\alpha }$$
where $O_{2,\mathrm{gas}}$ is the molecular oxygen adsorbed from the gas phase; $e^{-}$ is the elementary electrical charge withdrawn for the MOX conduction band; $O_{\beta ,\mathrm{surf}}^{-\alpha }$ is the ionic species chemisorbed over the oxide surface; β assumes the values of 1 or 2 for the molecular and atomic forms, respectively; while α is 1 or 2 for singularly and double-ionized ions.

Though Equation (1) provides an oversimplified view of the chemisorption process, it clearly shows the capability of the oxygen chemisorption process to modulate the electrical properties of the MOX semiconductor by modulating its density of charge carriers.

This mechanism is further exploited to detect molecules other than oxygen. Indeed, chemisorbed oxygen ions work as active species promoting the oxidation of other gas molecules (and the reduction of the oxide surface). For example, using carbon monoxide (CO) as prototypal molecule, its interaction with metal oxide gas sensors is typically described according to Equation (2) [35]:(2)$$\beta {\mathrm{CO}}_{\mathrm{gas}}^{}+O_{\beta ,\mathrm{surf}}^{-\alpha }\overset{k}{\to }\beta {\mathrm{CO}}_{2,\mathrm{gas}}^{}+\alpha e^{-}$$
where CO_gas_ is the CO molecule in the gas phase that adsorbs over the oxide surface; $O_{\beta ,\mathrm{surf}}^{-\alpha }$, e^−^, α and β are as above; ${\mathrm{CO}}_{2,\mathrm{gas}}^{}$ is the carbon dioxide molecule released back in air after the interaction between CO and the oxide semiconductor; and *k* is the reaction constant. However, despite its oversimplification, Equation (2) shows how the oxidation of the adsorbed CO molecule causes electrons previously withdrawn by oxygen chemisorption to be released back in the conduction band of the semiconductor.

As far as gases other than CO and O_2_ are concerned, reducing gases, such as, for example, ethanol and acetone, are oxidized by the interaction with chemisorbed oxygen following a reaction similar to the one reported in Equation (2) [46]. Oxidizing compounds, such as ozone and nitrogen dioxide, will oxidize the metal oxide surface through a reaction similar to Equation (1) [47].

The receptor function may feature gas specificity; hence, it is useful to address selectivity.

### 3.2. Transducer Function

As discussed in Section 2.3, the electrical transport thorough the sensitive layer can be decomposed into two contributions: one arising from elementary nanostructures and one from nanostructure–nanostructure junctions.

At microscopic level, junctions are characterized by an energy barrier that hinders the transport of charge carriers. In the case of nanoparticles, this barrier makes the junction conductance much lower than the particle conductance. As a consequence, the macroscopic resistance of traditional gas sensors based on nanoparticle networks, the so-called thick films, features junction-type characteristics, i.e., it is barrier-limited as described by Equation (3) [48]:(3)$$R\approx R_{j}=R_{0}\exp\left(\frac{E_{b}}{k_{B}T}\right)$$
where R_0_ is a pre-exponential factor with the dimension of a resistance, which also includes the geometrical details of the effective cross-sectional area for the thick film and its effective length; and E_b_ is the macroscopic energy barrier characterizing the system.

Until microscopic junctions can be considered almost equal to one another, E_b_ coincides with the barrier developed at individual microscopic junctions. If not, a statistical picture linking the microscopic and the macroscopic expressions of E_b_ should be used [49].

A schematic representation of the thick-film chemiresistor and the barrier arising at the nanoparticle–nanoparticle junction is provided in Figure 4. The sensing mechanism of thick films is typically explained in the literature on the basis of this scheme, Equation (3), and relating *E_b_* to the chemisorption and red-ox reactions described by Equations (1) and (2).

The chemisorption of oxygen over the MOX material creates acceptor-surface states, which in turn capture electrons from the semiconductor conduction band. As a consequence, charges accumulate at the particle–particle interface, developing an electric field that repels electrons from this region.

From a mathematical point of view, these effects are usually treated by adopting the following assumptions: (i) surface states capture electrons within a layer of width, *W*, which remains completely depleted by electrons; (ii) treating the surface of MOX crystallites as a flat plane, the mathematical problem can be solved in one dimension (ignoring the effects of the crystallite shape). Such a flat geometry is a good approximation for large grains, i.e., for grains with a diameter much larger than the depletion layer, D >> W. In this case (other situations will be considered in Section 3.4), in the inner part of the grain (up to W) the density of charge carriers is constant and equal to the bulk value of the semiconductor, N_0_. As detailed in Ref. [35], this abrupt distribution of carriers implies a parabolic bending of the semiconductor band structure, with the surface barrier reaching its maximum value E_b_ at the particle surface and restoring the unperturbed semiconductor properties in the core portion of the particle (between W and the center of the particle). The relationship between E_b_ and W is given by Equation (4), while the relationship between E_b_ and the density of surface sites N_S_ created by oxygen chemisorption is provided by Equation (5):(4)$$E_{b}=\frac{q^{2}N_{0}}{2\varepsilon }W^{2}=\frac{k_{B}T}{2}\frac{W^{2}}{\lambda_{D}^{2}}$$
(5)$$E_{b}=\frac{q^{2}N_{S}^{2}}{2{\varepsilon N}_{0}}$$

Here, q is the electron charge and ε is the dielectric constant of the semiconductor, $\lambda_{D}=\sqrt{{\varepsilon k}_{B}T/q^{2}N_{0}}$ is the Debye length, which is a characteristic length of the semiconductor expressing the distance over which mobile charge carriers screen a charge-induced perturbation. Its usefulness will be further discussed in Section 3.4 and Section 3.5, dedicated to size- and shape-effects.

A schematic representation of the depletion layer and the band bending is reported in Figure 4b.

Equation (3) combined with Equation (5) shows that the macroscopic resistance of the metal oxide layer depends exponentially from the squared density of surface states, whose value is modulated by the red-ox reactions with gaseous molecules. This relationship explains the high sensing capability of MOX chemiresistors and links this capability to the features of microscopic grain boundaries. The intensity of the sensing response, hereafter shortened as S, is also dominated by grain boundaries. S is calculated as the ratio between the resistances during exposure to the air background and air background with a given diluted amount of the target gas. For an n-type MOX exposed to reducing gases, S is expressed by Equation (6):(6)$$S=\frac{R_{\mathrm{air}}}{R_{\mathrm{gas}}}\approx \frac{R_{j,\mathrm{air}}}{R_{j,\mathrm{gas}}}=\exp\left(\frac{W_{\mathrm{air}}^{2}-W_{\mathrm{gas}}^{2}}{2\lambda_{D}^{2}}\right)=\exp\left(\frac{E_{b,\mathrm{air}}-E_{b,\mathrm{gas}}}{k_{B}T}\right)$$

Following the convention often adopted in gas sensing [51], which calculates the response in such a way that S > 1 during gas exposure, for n-type semiconductors exposed to oxidizing gases it is S = R_gas_/R_air_, while for p-type materials it is the opposite, i.e., S = R_gas_/R_air_ for reducing compounds and S = R_air_/R_gas_ for oxidizing ones.

As discussed in Section 2.3, the resistance of elementary nanowires R_nw_ may play a significant role both in single-nanowire and nanowire-mat-based devices. R_nw_ further depends on the interaction with gases and its relationship with the depletion layer developed at the nanowire surface can be retrieved based on geometrical considerations (conduction takes place only in the inner portion of the wire that is not depleted) [52]. This relationship is here reported in Equation (7) and it is the equivalent of Equation (3) for the nanowire body:(7)$$R_{\mathrm{nw}}=\frac{1}{{q\mu N}_{0}}\frac{L_{\mathrm{nw}}}{\pi {\left(\frac{D}{2}-W\right)}^{2}}=\frac{1}{{q\mu N}_{0}}\frac{L_{\mathrm{nw}}}{\pi {\left(\frac{D}{2}-2\sqrt{\frac{{\varepsilon E}_{b}}{q^{2}N_{0}}}\right)}^{2}}$$

L_nw_ is the nanowire length and μ is the bulk mobility of the metal oxide material.

Nanowire–nanowire junctions are typically modeled as grain boundaries of thick film materials, i.e., using Equation (3). For the nanowire body and the nanowire–nanowire junctions, E_b_ depends on surface states and the depletion layer, as described by Equations (4) and (5).

For an n-type MOX exposed to a reducing gas, the sensing response of the nanowire body then becomes:(8)$$S=\frac{R_{\mathrm{nw},\mathrm{air}}}{R_{\mathrm{nw},\mathrm{gas}}}=\frac{{\left(\frac{D}{2}-W_{\mathrm{gas}}\right)}^{2}}{{\left(\frac{D}{2}-W_{\mathrm{air}}\right)}^{2}}=\frac{{\left(\frac{D}{2}-2\sqrt{\frac{{\varepsilon E}_{b,\mathrm{gas}}}{q^{2}N_{0}}}\right)}^{2}}{{\left(\frac{D}{2}-2\sqrt{\frac{{\varepsilon E}_{b,\mathrm{air}}}{q^{2}N_{0}}}\right)}^{2}}$$

### 3.3. Utility Factor

With reference to Equation (2), an efficient receptor function implies a large k, which in turn means a large consumption of the target gas. If the sensing layer is thick, only the outermost portion of the sensitive film is exposed to the nominal concentration of the target gas, whose amount rapidly decreases proceeding toward the substrate.

These phenomena have been mathematically analyzed in terms of diffusion and surface reaction rates, obtaining the following expression for the gas concentration profile C as a function of the layer thickness, z [53]:(9)$$C\left(z\right)=C_{0}\frac{\cosh\left(\left(L-z\right)\sqrt{k/D_{k}}\right)}{\cosh\left(L\sqrt{k/D_{k}}\right)}$$

In Equation (9), L is the thickness of the sensitive film; C_0_ is the gas concentration in air and at the outermost layer of the sensitive film; and C_0_ = C(z = 0), $D_{k}=\frac{4r_{p}}{3}\sqrt{\frac{2\mathrm{RT}}{\pi M}}$ is the Knudsen diffusion coefficient, which depends on the pore radius r_p_, the molecular mass M, the working temperature T and the gas constant R.

This has been a leading concept in the development of several sensing layers and its optimization was identified as one of the key factors in some very remarkable achievements. Some examples will be discussed in Section 4.

### 3.4. Size Effects

The beneficial effects arising from a reduced diameter of elementary nanostructures is among the widest-acknowledged results in gas sensing. It has been widely reported in experimental works [44,54], and theoretical papers have explained its relationship with the optimization of the transducer function at the level of individual nanostructures [55,56,57].

Depending on the ratio between W and the diameter D of the elementary nanostructure, the electrical and sensing properties enter in different regimes. Among these, only the extreme cases admit an analytical solution, namely (i) D >> W and (ii) D ≤ λ_D_ (which also means D < W, according to Equation (4)). The intermediate conditions need numerical solutions [57].

In the first case, as evident from Equations (4) and (5) and from Figure 4, the interaction with gases alters the electrical properties of a given metal oxide only within a surface layer of thickness W. In the region beyond W, the properties of the material are insensitive to interaction with gases.

If the diameter of the nanostructure is further reduced below the extent of W, till the size of λ_D_, the material enters into the second regime, in which grains are fully depleted from electrons (apart those that are thermally promoted into the conduction band). In this condition, the barrier height at junctions is lower than the thermal energy (flat band condition). Phenomena described in Section 3.1 modulate the conductance of the semiconductor by modulating the position of its Fermi level, and the resulting transducer function is even more efficient than the one described by Equations (4) and (5). In particular, being D < W, D becomes the relevant characteristic length in the response intensity. More specifically, the response S increases with D^−1/2^ and with D^−1^ for reducing and oxidizing gases, respectively. The complete mathematical description of these regimes can be found in dedicated papers [55,56].

### 3.5. Shape Effects

Equation (4) is valid for plate-shaped crystallites or for other shapes in the case of large grains (D >> λ_D_), so that the surface of the crystallite can be considered almost flat. If grains are not large, as is often the case in nanostructures, shape effects apply to the E_b_ = E_b_(W) relationship modifying it with respect to Equation (4). As already introduced in Section 3.4, only a few regimes can be solved analytically, and this also holds for shape effects [57,58]. Despite such intrinsic difficulties, the proposed models and simulations agree that the spherical shape is the most effective to enhance the transduction mechanism [55,56,58]. In this sense, it is useful to compare two crystallites having the same diameter and electronic properties, undergoing the same interactions with gases, in particular oxygen, but featuring a spherical and cylindrical shape, respectively. Theories predict that the depletion layer is larger in the spherical crystallite than in the cylindrical one and that the former enters in the full-depletion regime earlier (for lower oxygen concentrations in the atmosphere) than the latter [55,56,58].

Moreover, spatially-resolved scanning tunneling spectroscopy (STS) experiments have shown that oxygen adsorption occurs preferably at grain boundaries and to a lesser extent over the crystallite surface [59]. This will reasonably have important implication in the comparison between nanowires and nanoparticles, since nanowire networks intrinsically feature a much lower density of junctions than thick films composed by nanoparticles. From this point of view, the spherical shape emerges once again as potentially more effective than the wire shape.

## 4. Approaches Adopted to Control the Sensing Properties of Gas Sensors

This section reports an overview of different strategies reported in the literature to exploit the morphological and structural features of MOX nanostructures to emphasize and tune their sensing capabilities.

Considering the vastness of the literature about nanowire chemiresistors, a complete overview of the field is out of the scope of the present work. Results will be shown referring mainly to SnO_2_, ZnO and WO_3_, which have been suitable benchmark materials to compare nanowires and nanoparticles since the early times of nanowire literature.

### 4.1. Porosity (Utility Factor) and Network Effects

As discussed in Section 3, small grains and large pores are fundamental features to optimize, respectively, the transduction mechanism and the efficiency of gas diffusion through the whole volume of the sensing layer.

Concerning traditional thick films, it has often been observed that these two morphological features may conflict one another, with small nanoparticles often implying small pores [60]. To solve this issue, several authors focused their work on the development of hierarchical nanostructures. These exploit nanoparticles organized in μm-sized assemblies that are further distributed in a disordered network connecting the electrodes [60]. Hollow spheres [61,62,63], fibers [64,65,66] and hollow fibers [67,68] are some popular examples.

Nanowire networks, on the other hand, intrinsically offer the possibility to merge these two requirements thanks to their nm-sized diameter and elongated shape, which often result in large voids, allowing an efficient gas diffusion.

Some remarkable results obtained through the optimization of these parameters will be reported in the following part of this section and will be summarized at the end of it in Table 1.

For example, Kida et al. compared the sensing properties of different SnO_2_ nanowire networks with varying length-to-diameter aspect ratios, including nanocubes with a size of ≈13 nm and nanorods with a diameter and length of ≈25 nm and ≈500 nm, respectively [69]. These nanostructures revealed excellent sensitivity to both ethanol and H_2_, demonstrating a very effective structure for the optimization of the transducer function and the utility factor. Concerning ethanol, a recent review highlighted that the response of these nanorods, ≈10^5^ to an ethanol concentration of 100 parts per million (ppm), emerged as the largest response reported in the literature among more than recent 80 papers about chemiresistors based on pure SnO_2_. In addition, nanocubes were identified as remarkable outliers in such a review [70]. The situation is similar for H_2_: these nanostructures revealed a very high response, at least comparable with state-of-the-art nanostructures such as thick films and hierarchical nanostructures with very thin, fiber-like shapes [71,72]. In addition to the excellent sensitivity in a general sense, the authors observed that size effects are emphasized with small molecules, in particular H_2_, while for larger molecules such as ethanol, the nanorods exhibited the best performance despite their diameter being larger than the nanocube size. This was ascribed to diffusion phenomena. For ethanol, this could take place in an effective manner only in the case of nanorods, whose length allowed for a larger porosity. Differently, for H_2_, the lower porosity featured by the nanocube layer was sufficient for an efficient diffusion of such a small molecule, thus emphasizing the beneficial size effects of the small nanocubes.

Other remarkable ethanol responses have been reported in the literature exploiting hierarchical assemblies of elementary nanowires. In this sense, it is worth mentioning the work by Firooz et al. that reports the response to 300 ppm of ethanol increasing from 1600 to 4000 when the structure is changed from a disordered network of nanowires to flower-like assemblies of nanowires [73]. These values are competitive with the largest responses recorded with nanoparticle-based devices, including thick films and hierarchical nanospheres with trimodal porosity [71,74,75].

In addition to the intrinsic porous structure, nanowire networks feature the presence of nanowire–nanowire junctions, which have often been invoked as key elements for the improvement of nanowire-based chemiresistors. This reasonably stems from the more efficient transduction mechanism of junctions with respect to the nanowire body (exponential vs. quadratic response, as depicted in Equations (6) and (8)), and to the preferential adsorption at grain boundaries discussed in Section 3.5. for example, junctions have been proposed to explain the remarkable responses to NO_2_ recorded at room temperature with In_2_O_3_ nanowire networks, in contrast with the much lower responses obtained with chemiresistors exploiting the single-nanowire configuration [76]. In particular, it is worth noting that the mentioned nanowire network revealed the ability to respond to NO_2_ concentrations lower than 50 parts per billion (ppb), which is the threshold limit for outdoor applications and is often used as a benchmark for sensing technologies, including thick films and hierarchical nanostructures [77,78,79]. On the other hand, such outstanding room-temperature sensitivity was obtained, exploiting an almost irreversible interaction with NO_2_, which required UV illumination to quickly restore the baseline.

Other remarkable NO_2_ responses at the level of 50 ppb and below were achieved, exploiting hierarchical WO_3_ nanostructures [80]. In this case, the devices exploited an embedded heater to work at the temperature of 300 °C, which ensured an effective compromise between intense response and acceptable response and recovery times.

Though several papers report the abundance of junctions as beneficial for nanowire chemiresistors [25,76,81,82], some works have recently pointed out the need for controlling the density of the nanowire network to achieve an optimal compromise between the density of junctions and the size of pores, which have opposite dependencies from the network density [83,84].

**Table 1 sensors-22-03351-t001:** Chemiresistors based on pure (neither doped nor functionalized) metal oxide (MOX) nanoparticles and nanowires. The response intensity S is calculated as S = R_gas_/R_air_ for NO_2_ and as S = R_air_/R_gas_ for other compounds at the sensor temperature T. Gas concentrations are expressed in parts per million (ppm) or parts per billion (ppb), and, in the T column, RT stands for ‘room temperature’.

MOX, Morphology	T (°C)	Gas, Concentration	S	Ref.
SnO_2_, nanowire network	250	Ethanol, 100 ppm	10^5^	[69]
SnO_2_, nanowire network	300	H_2_, 200 ppm	200	[69]
SnO_2_, nanocubes network	250	Ethanol, 100 ppm	6000	[69]
SnO_2_, nanocubes network	300	H_2_, 200 ppm	270	[69]
SnO_2_, nanowire network	300	Ethanol, 300 ppm	1600	[73]
SnO_2_, hierarchical flower-like assemblies of nanowires	275	Ethanol, 300 ppm	4000	[73]
SnO_2_, thick films	300	Ethanol, 100 ppm	1520	[71]
SnO_2_, thick films	300	H_2_, 200 ppm	87	[71]
SnO_2_, thick films	300	Ethanol, 100 ppm	2400	[74]
SnO_2_, hierarchical nanospheres of nanoparticles	400	Ethanol, 5 ppm	316	[75]
SnO_2_, hierarchical fibers of nanoparticles	250	H_2_, 100 ppm	25	[72]
SnO_2_, hierarchical fibers of nanoparticles	150	NO_2_, 125 ppb	90	[78]
WO_3_, 3D hierarchical assembly of nanowires	300	NO_2_, 50 ppb	6	[80]
WO_3_, thick film	300	NO_2_, 50 ppb	1.5	[77]
WO_3_, nanolamellae	200	NO_2_, 200 ppb	70	[79]
In_2_O_3_, nanowire network	RT	NO_2_, 50 ppb	2	[76]

### 4.2. Surface Termination

As discussed in Section 2.1, the surface termination of nanowires with well-defined crystalline planes is one of its unique features with respect to traditional nanoparticles. Indeed, different facets exhibit different densities of atomic edges, steps and unsaturated coordination sites that are all relevant for the interaction with gases [85,86]. Low-index surfaces of macroscopic SnO_2_ single crystals have been widely studied in surface science, showing their different behavior with respect to oxygen chemisorption [85]. Similarly, broad evidence has been collected demonstrating the surface-termination dependence of the interaction between macroscopic TiO_2_ single crystals and gaseous molecules [86].

Atomically resolved scanning tunneling microscopy studies have been applied to SnO_2_ nanowires pre- and post-surface-oxidation and surface-reduction treatments. Results have shown a similar ordering of surface atoms for these nanocrystals and macroscopic crystals typically studied in traditional surface science, providing an important conceptual link between the two disciplines [87].

More recently, attention has been dedicated to high-index surfaces, which are expected to be more suitable than low-energy ones for gas sensing owing to their richness in atomic edges, steps and unsaturated coordination sites. This has been confirmed both computationally and experimentally. Density functional theory (DFT) calculations confirmed that the exothermic oxygen chemisorption over SnO_2_ surfaces is more favorable (larger energy reduction) for high-energy facets such as (221) than for low-energy ones, such as (110) [88,89,90]. From an experimental point of view, the comparison between the gas sensing properties of low- and high-index surfaces has been realized through the synthesis of nanopolyhedra exposing high-energy facets and the synthesis of elongated nanopolyhedra, laterally bonded by low-energy facets, using SnO_2_ as example material [91]. In this way, the control over the length of the elementary nanostructures allowed for tuning of the balance between the areas of high- and low-energy planes, and in turn, their relative contributions to the sensing response. Similar improvements have also been recorded for other oxides, including Fe_2_O_3_ [92], Cu_2_O [90], NiO [30] and TiO_2_ [93]. In 2018, these results were analyzed as a whole. It was concluded that in most of cases, facets revealed more effectiveness than surface area in the enhancement of the sensor response in a broad sense, i.e., the improvement was observed toward different gases [94].

In other papers, the surface termination was proposed to explain the observed differences about the partial selectivity exhibited by nanowires and nanoparticles. For example, comparing SnO_2_ nanowire and nanoparticle networks, it was observed that none of the two morphologies could be indicated as more sensitive than the other in an absolute way, but these considerations are gas-dependent. As reported in Figure 5, nanowires were found to be more sensitive to compounds such as acetone, dimethyl methylphosphonate (DMMP) and dipropylene glycol monomethyl ether (DPGME), while nanoparticles were more sensitive to CO and NH_3_ [95]. Similar results were also obtained with In_2_O_3_ nanowires and nanoparticles [95]. Differences were also observed in the case of WO_3_, for which nanoparticles and nanorods exhibited similar responses to ammonia; while in the case of ethanol and acetone, the response of nanoparticles was about one order of magnitude larger than the response of nanorods [25].

### 4.3. Doped Nanostructures

In this paper, the term ‘doping’ will be intended according to its meaning from the field of semiconductors, in which a dopant is an additive element introduced in the lattice of the host material as an interstitial or substitutional atom/ion [96]. Those cases in which the additive element is deposited in the form of cluster over the supporting MOX are referred to as surface functionalization and will be treated in Section 4.4.

In the field of thin- and thick-film chemiresistors, the dispersion of dopants has been widely used to address several objectives: (i) to increase the thermal stability of the film micro/nanostructure by hindering grain-coarsening phenomena; (ii) to decrease the electrical resistivity of the film; (iii) to control its Debye length and the space-charge-layer depth; (iv) to tailor the response and the partial selectivity of a given material according to specific requirements.

Between these objectives, the former is probably the most shape-dependent. For thin and thick films composed by more or less compact aggregates of spherical nanostructures, grain coarsening was recognized as an important limiting factor for the long-time stability of MOX-based chemiresistors [97]. Electron microscopy investigations revealed that these phenomena occur at grain boundaries and involve the rearrangement of atom ordering at these interfaces. Reordering is such that larger grains coarsen by subtracting atoms from smaller grains [98]. Considering the importance of the grain size for the sensing properties of MOX, these microscopic phenomena affect the macroscopic properties of the device, such as its baseline and sensing response. To suppress these microstructural drifts, dopants have been widely used in the field of ceramic materials. These additives act as blocking elements for grain-boundary migration and phase transition [97,99].

As for nanowires, their elongated single-crystalline structure has already been proposed as a possible solution for such microstructural drifts in the first publication about nanowire-based gas sensors [2]. This is intrinsic for devices based on a single nanowire or aligned nanowires owing to the total absence of grain boundaries. For nanowire networks, which feature the presence of nanowire–nanowire junctions, an increased stability with respect to nanoparticles was reported after a study lasting about 1 month [100].

From this point of view, Ga_2_O_3_ is of particular interest. It has been largely studied as polycrystalline-film gas sensor owing to its thermal and chemical stability, which make it an appealing material for gas sensing [101,102]. In its pristine form (not intentionally doped), grains about 20 nm large remain stable at the temperature of 900 °C [103], while for pristine SnO_2_ the stable diameter is of the order of 100 nm [104]. Ga_2_O_3_ nanowires have also been investigated for H_2_ and volatile organic-compound sensing [105,106].

In addition to grain coarsening, doping also affects all the other aforementioned features of the device. Doped materials typically exhibit a baseline that is different from that of the pristine host. From this viewpoint, dopants are often used to decrease the baseline of materials that exhibit a large resistivity. Indeed, it may happen to deal with materials/sensors whose baselines approach or even exceed the GOhm [97,107,108]. The electrical signals of these devices are easily read by laboratory equipment but may be hard to read with cheap commercial electronics. Research in readout systems is studying innovative and cheap approaches suitable to read resistances spanning a broad range [109,110]. Nonetheless, avoiding very large values may be an appealing feature for easy and effective device exploitation [111]. Doping of TiO_2_ is a typical example: In its pure form, titania features the presence of oxygen vacancies but its resistivity remains quite large, and dopants such as Nb are often used to mitigate this drawback owing to the donor properties of Nb in TiO_2_. Nb-doped TiO_2_ has been studied both in the form of thick films [97], nanorods [112] and hierarchical structures such as nanotubes [113]. Ga_2_O_3_ is also highly resistive at the typical gas-sensing temperatures (below 600 °C), and dopants such as, for example, Si and Sn, are often used to increase the charge-carrier density [114,115].

Altering the baseline of the host material through the introduction of dopants also affects the transduction mechanism through the modulation of the Debye length. According to semiconductor theory (see Section 3.1), the increase in charge carriers implies shorter Debye length (and space-charge layer), which in turn implies a reduced sensor response. Generally speaking, a balance between an easy-to-read baseline and an effective Debye length should be considered.

In some cases, doping may also cause the material to switch from n- to p-type or vice-versa. Using TiO_2_ as example, this is the case for Cr doping, which enters in the TiO_2_ lattice as electron acceptor, hence introducing holes. P-type Cr-doped TiO_2_ has been synthesized introducing Cr at the concentration of ca 8% at. [107].

In addition, dopants also affect the receptor function of the host material and can be exploited to tune the partial selectivity of MOX. For example, concerning Ga_2_O_3_, Sb doping enhances its response to O_2_, while Na-K co-doping improves its sensitivity to humidity [116,117]. Zn doping in In_2_O_3_ nanowires was reported to modify the response spectrum of the guest oxide by decreasing its response to NO_2_ and increasing the response to reducing gases, more specifically to H_2_, CO, ethanol and acetone [118]. A general increase in the response toward reducing compounds was also observed for Zn-doped In_2_O_3_ hierarchical nanospheres composed by spherical crystallites, with emphasis on the enhancement of the response to triethylamine [119]. Both for nanowires and other nanostructures, the effect of Zn doping on In_2_O_3_-sensing properties is usually ascribed to the formation of point defects, such as Zn interstitials, oxygen and In vacancies in the host matrix [118,120]. In some papers, Zn has also been proposed to stimulate the phase transition of the host oxide. This is the case, for example, of the In_2_O_3_ transition from body-centered cubic (bcc) to rhombohedral (rh). Introducing Zn in single-phase pristine bcc-In_2_O_3_, the material changed into a polycrystalline mixture of bcc/rh, with the portion of the rh phase increasing with the increase in the dopant [120]. This structural modification was accompanied by a morphological evolution from single-crystalline nanocubes to hierarchical polycrystalline nanoflowers. The joint structural and morphological modification implied more point and extended defects, including bcc-rh interfaces, and an increased surface area, which were identified as the key features underlying the increased sensitivity of doped samples to NO_2_. Morphological changes promoted by Zn doping were also observed in hierarchical spheres composed by SnO_2_ nanorods, which turned to assemblies of sheet-shaped crystallites. Zn-doped SnO_2_ nanostructures revealed more sensitivity in general toward reducing gases, including ethanol, glycol and acetone [121].

### 4.4. Inorganic Heterostructures

The functionalization of MOX surfaces with inorganic additives is probably the widest-used method to improve and tune the properties of MOX chemosensors. Both metallic nanoparticles and MOX nanoparticles have been widely investigated as additives to MOX thick films and have also further been applied to MOX nanowires. Over the years, several review papers have been specifically dedicated to this topic. In the past, these were mainly concerned with chemiresistors based on thick and thin films [122], while in more recent years, results obtained with other morphologies such as nanowires, nanosheets and hierarchical structures have also been included [123,124].

#### 4.4.1. Functionalization with Metallic Nanoparticles

As for metallic additives, their effect (often termed sensitization) on gas-sensing properties occurs through two mechanisms: electronic and chemical [125]. The former arises from the different work functions of the metal and the MOX semiconductor (*ϕ_m_* and *ϕ_s_*). It is generally explained starting from the ideal situation of the two isolated materials that are further brought into contact. Referring to an isolated n-type MOX, its Fermi level lies inside the bandgap, close to the conduction band owing to oxygen vacancies. As a result, it is usually observed that *ϕ_m_* > *ϕ_s_*, meaning that it is easier to extract electrons from the semiconductor than from the metal. When the two systems are coupled, the Fermi levels’ alignment implies the transfer of electrons from the semiconductor to the metal, resulting in a depletion layer in the MOX side of the interface [35,126]. This effect adds up with the depletion layer induced by oxygen chemisorption described in Section 3 [125]. A schematic representation of the electronic sensitization is shown in Figure 6a. The second mechanism stems from the catalytic properties of nanosized metallic particles, which are able to dissociate molecules into byproducts that are further spilled over the supporting oxide (spill-over effect) facilitating the overall response of the composite material [122]. For chemosensing, the dissociation of molecular oxygen into reactive O^−^ ions promoted by metals such as Au and Pd is of particular relevance [127,128,129]. It acts by increasing both the density of chemisorbed oxygen ions and the depletion-layer extension in the surrounding of the metallic nanoparticle, as schematically shown in Figure 6b.

In addition to oxygen, other molecules are also directly spilled over by suitable metallic nanoparticles. A widely known example is the spill-over of H_2_ molecules promoted by Pd and Pt, which has been largely exploited in thick-film technology [122] as well as with nanowire chemiresistors, both in the single-nanowire [129] and nanowire-network configurations [18,69].

In other papers, the synergy between the intrinsic catalytic activity of metallic nanoparticles and the increased density of reactive oxygen arising from oxygen spillover is proposed as the main reason for the increased response. This is the case of ethanol responses enhanced by Au or Pt nanoparticles. To cite a few examples, Au functionalization was employed with SnO_2_ layers, both in the form of thick films [130] and nanowires [131]; Pt was used with SnO_2_ hollow spheres [132] and networks of SnO_2_ nanorods [133]. The same combination of metal-promoted effects was also proposed to explain the improved response to H_2_S observed for several Au-supported MOX, including WO_3_ nanorods [134] and nanoparticles [135].

A summary of the numerical responses recorded with these materials, both in their pristine and functionalized form, is reported in Table 2. In addition to the improved response intensity toward the target gas, it is worth mentioning the beneficial effect about the optimal working temperature, which is often lowered by the functionalization with metallic nanoparticles.

In recent years, bimetal nanoparticles composed by alloyed metals have attracted large attention owing to their coupling with the supporting oxide, which is different from the coupling of the respective monometallic components. For example, the AuPd system has been studied by different research groups using different supporting metal oxides. Considering SnO_2_ thick films as support, the different balance between oxygen spillover and electronic sensitization has been reported for Au, Pd and AuPd nanoparticles [128]. Despite these insightful results, the effects of the bimetallic functionalization with respect to the functionalization with individual metal is still in an early stage. Different results have been published so far, without reaching a comprehensive, uniform picture of the involved phenomena. For example, enhanced performances have been reported for AuPd-functionalized SnO_2_ thick films to a broad range of compounds, including CO, CH_4_ and NH_3_, [136]. However, other papers reported AuPd functionalization as less effective for CO, ethanol and CH_4_ with respect to the pristine and single-metal-functionalized SnO_2_ [128]. Morphologies other than thick films have also been considered for this kind of functionalization. Some examples are SnO_2_ flower-like hierarchical assemblies of nanosheets, which showed an increased sensitivity to several compounds, including formaldehyde and acetone [137]; SnO_2_ hollow spheres, for which the effectiveness of the functionalization was proven against dimethyl disulfide; and WO_3_ nanowires, tested against butanol [138].

Before concluding this paragraph, it is worth mentioning another approach to exploit the metal–semiconductor interface for gas sensing, which was proposed by Wei et al. [139] working with a single-nanowire device. In this case, the metallic structure was the electrode, which formed a reverse biased Pt-ZnO junction in the single-nanowire device. Based on the discussion reported in Section 2.3, though such an interface is also present in devices based on MOX thick films and nanowire networks, it is particularly meaningful in the single-nanowire device, for which such a junction may dominate the overall device resistance. In particular, Wei et al. [139] proposed a sensing mechanism based on the modulation of the Schottky barrier E_ms_ at the metal–oxide interface induced by gas adsorption:(10)$$R\approx R_{\mathrm{cr}}+R_{\mathrm{nw}}\approx R_{\mathrm{cr}}\propto \exp\left(\frac{E_{\mathrm{ms}}}{k_{B}T}\right)\;\exp\left(-\frac{{\mathrm{qV}}_{\mathrm{cr}}}{k_{B}T}\right)$$
(11)$$S\approx \frac{R_{\mathrm{cr},\mathrm{gas}}}{R_{\mathrm{cr},\mathrm{air}}}=\exp\left(\frac{E_{\mathrm{ms},\mathrm{gas}}-E_{\mathrm{ms},\mathrm{air}}}{k_{B}T}\right)\;$$

V_cr_ is the voltage drop over the reversely biased metal–oxide interface and other symbols are defined as above. Although in this case the metal is not dispersed in the form of nanoparticles—hence neither the electronic nor the catalytic sensitization are optimized—its transduction mechanism features an exponential dependence from E_ms_, which is more efficient than the quadratic form of the nanowire body described by Equation (8).

**Table 2 sensors-22-03351-t002:** Chemiresistors based on metal oxide (MOX) nanostructures functionalized with metallic nanoparticles. The response intensity S is calculated as S = R_air_/R_gas_ at the sensor temperature T. In case of no response at this temperature, ‘no resp.’ is reported in the S column. Gas concentrations are expressed in parts per million (ppm).

Supporting MOX, Morphology	Functionalization	T (°C)	Gas, Concentration	S	Ref.
SnO_2_, nanowire network	--	250	Ethanol, 100 ppm	10^5^	[69]
SnO_2_, nanowire network	Pd	250	Ethanol, 100 ppm	1.1 × 10^5^	[69]
SnO_2_, nanowire network	--	300	H_2_, 200 ppm	200	[69]
SnO_2_, nanowire network	Pd	250	H_2_, 200 ppm	800	[69]
SnO_2_, nanocubes network	--	250	Ethanol, 100 ppm	6000	[69]
SnO_2_, nanocubes network	Pd	250	Ethanol, 100 ppm	6000	[69]
SnO_2_, nanocubes network	--	300	H_2_, 200 ppm	270	[69]
SnO_2_, nanocubes network	Pd	250	H_2_, 200 ppm	300	[69]
SnO_2_, nanowire network	--	150	H_2_, 40 ppm	no resp.	[18]
SnO_2_, nanowire network	Pd	150	H_2_, 40 ppm	3	[18]
SnO_2_, single nanowire	--	100	H_2_, 1 ppm	no resp.	[129]
SnO_2_, single nanowire	Pd	100	H_2_, 1 ppm	5	[129]
SnO_2_, thick film	--	270	Ethanol, 200 ppm	28	[130]
SnO_2_, thick film	Au	220	Ethanol, 200 ppm	128	[130]
SnO_2_, hollow spheres	--	325	Ethanol, 5 ppm	95	[132]
SnO_2_, hollow spheres	Pt	325	Ethanol, 5 ppm	1400	[132]
SnO_2_, nanorod network	Pt	300	Ethanol, 200 ppm	40	[133]
WO_3_, nanorod network	--	350	H_2_S, 1 ppm	4	[134]
WO_3_, nanorod network	Au	350	H_2_S, 1 ppm	100	[134]
WO_3_, thick film	--	300	H_2_S, 1 ppm	3	[135]
WO_3_, thick film	Au	300	H_2_S, 1 ppm	7	[135]
SnO_2_, thick film	--	300	CO, 50 ppm	10	[128]
SnO_2_, thick film	Au	300	CO, 50 ppm	100	[128]
SnO_2_, thick film	Pd	300	CO, 50 ppm	100	[128]
SnO_2_, thick film	AuPd	300	CO, 50 ppm	2.5	[128]
SnO_2_, thick film	--	300	Ethanol, 10 ppm	50	[128]
SnO_2_, thick film	Au	300	Ethanol, 10 ppm	500	[128]
SnO_2_, thick film	Pd	300	Ethanol, 10 ppm	150	[128]
SnO_2_, thick film	AuPd	300	Ethanol, 10 ppm	40	[128]
SnO_2_, thick film	--	300	CH_4_, 1000 ppm	12	[128]
SnO_2_, thick film	Au	300	CH_4_, 1000 ppm	30	[128]
SnO_2_, thick film	Pd	300	CH_4_, 1000 ppm	90	[128]
SnO_2_, thick film	AuPd	300	CH_4_, 1000 ppm	12	[128]
SnO_2_, thick film	--	350	CO, 20 ppm	3	[136]
SnO_2_, thick film	Au	225	CO, 20 ppm	5	[136]
SnO_2_, thick film	Pd	100	CO, 20 ppm	3	[136]
SnO_2_, thick film	AuPd	350	CO, 20 ppm	9	[136]
SnO_2_, thick film	--	500	CH_4_, 50 ppm	3	[136]
SnO_2_, thick film	Au	450	CH_4_, 50 ppm	4.5	[136]
SnO_2_, thick film	Pd	450	CH_4_, 50 ppm	3	[136]
SnO_2_, thick film	AuPd	400	CH_4_, 50 ppm	6.5	[136]
SnO_2_, thick film	--	375	NH_3_, 10 ppm	2	[136]
SnO_2_, thick film	Au	350	NH_3_, 10 ppm	4	[136]
SnO_2_, thick film	Pd	350	NH_3_, 10 ppm	2	[136]
SnO_2_, thick film	AuPd	350	NH_3_, 10 ppm	6	[136]
SnO_2_, nanosheet network	--	300	Acetone, 50 ppm	20	[137]
SnO_2_, nanosheet network	Au	275	Acetone, 50 ppm	80	[137]
SnO_2_, nanosheet network	Pd	250	Acetone, 50 ppm	40	[137]
SnO_2_, nanosheet network	AuPd	250	Acetone, 50 ppm	110	[137]
WO_3_, nanowire network	--	300	n-butanol, 100 ppm	26	[138]
WO_3_, nanowire network	Pd	200	n-butanol, 100 ppm	69	[138]
WO_3_, nanowire network	AuPd	200	n-butanol, 100 ppm	93	[138]

#### 4.4.2. Functionalization with Metal Oxide Nanoparticles

Concerning the use of MOX nanostructures as catalysts, their beneficial effects arise both from the catalytic properties of the additive and from the formation of electrical junctions between the two MOX semiconductors.

CuO is a p-type semiconductor and the formation of p-n junctions at the interface with n-type MOX is often indicated as one of the key reasons for the observed enhanced sensitivity with respect to the pure n-type MOX. For example, this is the case of ethanol sensing with pristine and CuO-functionalized SnO_2_ hollow spheres [61]. In addition, CuO catalyst is particularly effective for H_2_S detection, which is probably its most popular use in gas sensing [140]. This is thanks to the suitability of CuO to react with H_2_S forming CuS according to the reaction $\mathrm{CuO}+H_{2}S\;\to \mathrm{CuS}+H_{2}O$, which is reversible in an oxygen-rich environment [141]. Considering the metallic character of CuS, the p-n junction formed at the CuO–MOX interface is turned into a metal–semiconductor junction upon H_2_S exposure, resulting in a very effective transduction mechanism. This mechanism is reported as being specific to H_2_S. Indeed, as shown in Table 3, large H_2_S response enhancements have been published for SnO_2_-CuO composites based both on nanoparticles and nanowires as supporting material, while weaker enhancements have been indicated for other compounds such as CO and NH_3_ [46,141].

In addition to p-n junctions, n-n junctions have been exploited. A remarkable example is given by SnO_2_ and ZnO, which are widely used as individual material in a variety of morphologies, but they are also used combined with one another. Hierarchical morphologies are also widely employed to combine the beneficial morphological and functionalization effects. Concerning nanoparticle-shaped crystallites, SnO_2_ nanofibers increased their response to ethanol and acetone by a factor of about three when decorated with ZnO nanoparticles [64]. This is thanks to the combined morphological effect, arising from the hierarchical structure, and the functionalization one. Porous opals with 1:1 SnO_2_−ZnO composition were also realized [142]. In this case, the response to acetone was increased with respect to the same morphology realized with a single component, SnO_2_ or ZnO, while for ethanol the composite material featured a response comparable with the one exhibited by pure SnO_2_ and larger than the pure ZnO response, hence also providing benefits in terms of partial selectivity.

Concerning nanowires, Zhao et al. studied the effect of SnO_2_-nanoparticle loading over the surface of ZnO nanowires for NO_2_ detection. Their results indicated that the functionalization improves the response intensity to NO_2_ by a factor of about six with respect to the performance of pristine ZnO [143]. NO_2_ was also the target gas for SnO_2_ nanowires cofunctionalized with ZnO nanoparticles and Pd nanoparticles, finding a nearly 3-times increase in the response of the composite material with respect to the base SnO_2_ nanowires [144].

**Table 3 sensors-22-03351-t003:** Chemiresistors based on metal oxide (MOX) nanostructures functionalized with metal oxide nanoparticles. The response intensity S is calculated as S = R_air_/R_gas_ for reducing gases and as S = R_gas_/R_air_ for NO_2_ at the sensor temperature (T). Gas concentrations are expressed in parts per million (ppm). In those cases in which information is not available from the original reference, the related cell reports ‘na’.

Supporting MOX, Morphology	Functionalization	T (°C)	Gas, Concentration	S	Ref.
SnO_2_, hollow spheres	--	300	Ethanol, 300 ppm	11	[61]
SnO_2_, hollow spheres	CuO	300	Ethanol, 300 ppm	35	[61]
SnO_2_, thick film	--	350	H_2_S, 2 ppm	100	[46]
SnO_2_, thick film	CuO	200	H_2_S, 2 ppm	600	[46]
SnO_2_, thick film	--	350	CO, 40 ppm	3	[46]
SnO_2_, thick film	CuO	350	CO, 40 ppm	4	[46]
SnO_2_, thick film	--	300	NH_3_, 20 ppm	2	[46]
SnO_2_, thick film	CuO	250	NH_3_, 20 ppm	2	[46]
SnO_2_, nanowire network	--	400	H_2_S, 2 ppm	8	[141]
SnO_2_, nanowire network	CuO	200	H_2_S, 2 ppm	3261	[141]
SnO_2_, nanowire network	--	na	CO, 50 ppm	7	[141]
SnO_2_, nanowire network	CuO	na	CO, 50 ppm	7	[141]
SnO_2_, nanowire network	--	na	NH_3_, 17 ppm	4	[141]
SnO_2_, nanowire network	CuO	na	NH_3_, 17 ppm	4	[141]
SnO_2_, single nanowire	--	250	H_2_S, 10 ppm	1.5	[141]
SnO_2_, single nanowire	CuO	250	H_2_S, 10 ppm	26	[141]
SnO_2_, porous fiber	--	260	Ethanol, 100 ppm	120	[64]
SnO_2_, porous fiber	ZnO	260	Ethanol, 100 ppm	360	[64]
SnO_2_, porous fiber	--	260	Acetone, 100 ppm	10	[64]
SnO_2_, porous fiber	ZnO	260	Acetone, 100 ppm	30	[64]
SnO_2_, porous opal	--	250	Acetone, 100 ppm	13	[142]
SnO_2_/ZnO, porous opal	--	275	Acetone, 100 ppm	45	[142]
ZnO, porous opal	--	350	Acetone, 100 ppm	17	[142]
SnO_2_, porous opal	--	250	Ethanol, 100 ppm	22	[142]
SnO_2_/ZnO, porous opal	--	250	Ethanol, 100 ppm	23	[142]
ZnO, porous opal	--	325	Ethanol, 100 ppm	12	[142]
SnO_2_, nanowire network	--	300	NO_2_, 5 ppm	2	[144]
SnO_2_, nanowire network	Pd	300	NO_2_, 5 ppm	4	[144]
SnO_2_, nanowire network	ZnO	300	NO_2_, 5 ppm	4	[144]
SnO_2_, nanowire network	ZnO + Pd	300	NO_2_, 5 ppm	6	[144]

#### 4.4.3. Core–Shell Nanostructures

The exploitation of the MOX/MOX interface is also at the basis of core–shell nanostructures, in which the inner nanowire material (core) is completely coated by the shell layer, which typically has a polycrystalline structure. An example is reported in Figure 7, with reference to a SnO_2_-ZnO core–shell sample. In this type of nanostructure, the outer layer is usually compact, and the composite material exposes the shell material to the gaseous environment. From the functional point of view, this is particularly evident in the case of core–shell nanostructures composed by a p-n (or n-p) couple of semiconductors, for which the sensing mechanism of the core is clearly distinguishable from that of the shell [145,146]. The transduction mechanism of the composite is strongly dependent on the core–shell electrical coupling, which in turn depends on the shell thickness. Investigations carried out with different nanowire-core/polycrystalline-shell systems, including CuO/ZnO [145], SnO_2_/Cu_2_O [146] and SnO_2_/ZnO [147], indicated that the sensing response is optimized for a shell thickness approaching the Debye length (λ_D_). The explanation of these results is typically carried out starting from the band structure of the two materials in contact with one another, assuming both materials are thick enough to restore the bulk properties outside the space-charge layer [145,147,148,149]. Figure 7 reports an example of this band structure using SnO_2_-ZnO as a reference system. Starting from this framework, thickness arguments can be further briefly discussed as follows [147]: decreasing the shell thickness increases the sensitivity of the shell material to the interaction with gases, according to space-charge-layer arguments similar to those described in Section 3.2. Nonetheless, as the shell is thinned, an increasing amount of electrical transport takes place in the core instead of in the shell, thus decreasing its coupling with gas interaction occurring at the external surface of the shell. These results have been reported to apply in general to reducing gases and to core–shell systems, whichever their n-n, p-n or n-p character [145,146,147]. Such a general improvement, with poor specificity among reducing compounds, has also been confirmed by other authors [149]. The Ga_2_O_3_/SnO_2_ system has also been exploited in the core–shell configuration, obtaining a reduction in the optimal temperature for ethanol detection with respect to the pristine Ga_2_O_3_ nanowire (400 vs. 600 °C) [150].

Joint functionalization with MOX and with metallic nanoparticles is also reported in the literature. For example, a partial specificity to triethylamine was obtained by further functionalizing the ZnO-SnO_2_ core–shell composite with Au nanoparticles, which allows to jointly exploit the gas specificity of the catalyst and beneficial effects of the core–shell structure on the transduction mechanism [149].

The sensing properties of these materials are resumed in Table 4 in terms of sensor response intensity S and type of sensing mechanism (n- vs. p-type).

#### 4.4.4. Hierarchical, Branched Nanostructures

An additional opportunity offered by nanowires, which does not find an equivalent in the nanoparticle case, is the possibility to grow branched heterostructures composed by a nanowire backbone, with other nanowires extending from its surface. This is a very interesting configuration because it allows at the same time for the exploitation of more phenomena described in the previous sections. An example of this morphology is shown in Figure 8a. The two main features of this particular type of hierarchical structure for gas sensing are: (i) the very open structure, which enhances the utility factor, and (ii) the MOX–MOX interface, which may be in the core–shell form or as dispersed branches growing out from the backbone nanowire, providing the functionalities already introduced in Section 4.4.3 and Section 4.4.2, respectively.

Using the SnO_2_-ZnO system as a case study, Tharsika et al. compared the performance of three different morphologies, namely the SnO_2_ nanowire network in its pristine form, the same base material in a core–shell configuration using a ZnO film coating (Figure 7) and in a hierarchical configuration in which branched ZnO nanorods grow out the ZnO shell. Results indicate that this latter configuration is the most effective both in terms of response intensity to different compounds and in terms of partial selectivity to ethanol [148]. The Cu_x_O-ZnO system has also been investigated as a branched structure for acetone sensing, obtaining a response about four times larger with respect to the individual nanowires [151]. In this case, ZnO branches are also not dispersed over the Cu_x_O backbone but directly emerge from a uniform ZnO shell coating the underlying Cu_x_O nanowire core. The response type indeed switches from p- to n-type as the backbone is coated by ZnO.

Branched nanowire heterostructures have also been exploited as hierarchical support for further functionalization with metallic or MOX nanoparticles. For example, the structure composed by SnO_2_ nanowire as backbone, dispersed Bi_2_O_3_ branches and Pt nanoparticles has been used to detect NO_2_ at sub-ppm levels [152]. The branched nanostructure (without Pt functionalization) revealed the best performance, with optimal performances reached at the working temperature of 300 °C owing to the SnO_2_-Bi_2_O_3_ interface exposed to gases. Benefits arising from the addition of Pt consisted in reducing the optimal temperature to 50 °C for an almost room-temperature sensing [152]. Other branched, ternary materials have also been studied for NO_2_ sensing. These are hierarchical structures composed by branched ZnO nanorods grown over SnO_2_ nanowire and further functionalized with either Au [153] or Cr_2_O_3_ [154] nanoparticles. As for the SnO_2_-ZnO-Au system, optimal performances were obtained at the working temperature of 300 °C, with the response of the ternary compound being about four times larger than the response of the branched nanostructure without Au nanoparticles and about six times the response of pristine SnO_2_ nanowires. The increase is even larger for the SnO_2_-ZnO-Cr_2_O_3_ system, its response to NO_2_ being about ten and twenty times the response of the branched SnO_2_-ZnO and the pure SnO_2_ nanowires. These results were ascribed to the synergic effects of the branched structure of SnO_2_-ZnO hierarchical support, their n-n interfaces and the effects of Au and Cr_2_O_3_, which act through the electronic and chemical sensitization mechanisms discussed in Section 4.4.1 [153,154].

Doping has also been used in conjunction with branched nanostructures. For example, branched SnO_2_ nanowires were grown over metallic Sb-doped SnO_2_, realizing a hierarchical structure characterized by semiconductor–metal interfaces. This system was used for ethanol sensing [155].

The response intensities S and the type of sensing mechanisms of the branched nanomaterials discussed in this section are compared with those of the respective backbone nanostructures in Table 5.

**Figure 8 sensors-22-03351-f008:**
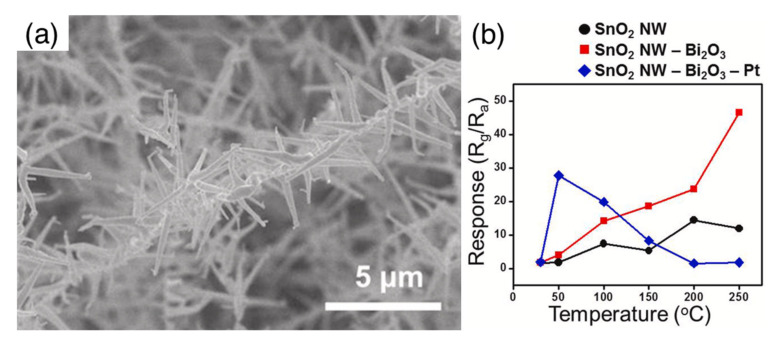
Branched hierarchical nanowires. SEM image of branched Bi_2_O_3_ nanowires grown over SnO_2_ nanowires (**a**). Sensor response to 1 ppm of NO_2_ vs. temperature for the branched Bi_2_O_3_–SnO_2_ nanowires functionalized with Pt nanoparticles, branched Bi_2_O_3_–SnO_2_ nanowires and the backbone SnO_2_ nanowires (**b**). Figure 8a,b are reprinted from [152], Copyright (2021) with permission from Elsevier.

### 4.5. Inorganic–Organic Heterostructures

In addition to inorganic materials, organic layers have also been considered to functionalize metal oxides. The goal is to exploit the variety of interactions offered by the organic chemistry to control the sensitivity and the partial selectivity of the composite material.

#### 4.5.1. Graphene and Related Materials

Nowadays, talking about organic functionalization immediately recalls graphene (G) and related materials such as graphene oxide (GO) and reduced graphene oxide (RGO). Indeed, several works have been published exploiting both nanoparticles, nanowires and hierarchical nanostructures coupled with these carbon-based materials. Similarly to other functional interfaces mentioned in the previous sections, the electrical coupling between the two materials is a fundamental concept to explain the gas-sensing properties of the composite. For 2D materials, the band structure is depicted through Dirac cones, with the Fermi level and the shape of cones strongly depending on defects and number of layers, as well as bending and corrugation that may affect the 2D film [156]. These parameters, in turn, exhibit a large dependence from the synthesis conditions of the carbon-based material and from the morphological features of the surface over which it is deposited/synthesized. As a consequence, the electrical properties of these materials are still the subject of study and a unified picture is still to be achieved, especially for those situations in which many of the aforementioned effects may be involved at the same time, as is the case in gas-sensor devices.

In addition, G, GO and RGO are intrinsically sensitive to gases, hence they may also directly contribute to the overall response of the device [157,158].

The result is a large number of parameters that may be tuned to control the sensing properties of the composite material.

An example of hybrid material composed by ZnO nanorods and RGO sheets is reported in Figure 9a. At the interface between the two materials, electrical coupling gives rise to the formation of p-n inorganic–organic junctions, which are adopted to interpret the electrical and sensing properties of the composite. Figure 9b reports the schematic representation of the band structure of the RGO-ZnO interface in three conditions: the two separated materials, the coupled materials exposed to air and the coupled materials exposed to NO_2_. Considering the work functions of RGO (≈4.4–5 eV) and of ZnO (≈4.2–4.3 eV), the alignment of the Fermi levels implies electrons transferring from ZnO to RGO, hence a depletion layer extending inside the ZnO nanorods [159]. Already at low RGO concentrations, this material was proposed to facilitate the electrical transport between connected nanorods owing to the reduced resistance of RGO with respect to the ZnO-ZnO interface. On the other hand, considering the low RGO concentration in the optimal device, the sensing mechanism was interpreted as the NO_2_ molecules adsorbing mainly on ZnO. The proposed transduction mechanism involved the consequent downshift of the ZnO Fermi level, which in turn resulted in the electrons’ withdrawal from the RGO phase and the hole concentration increase therein. This explained the experimentally observed p-type response (electrical resistance decrease upon gas exposure) of the system at room temperature [159]. On the other hand, n-type sensing is also reported in the literature for the RGO–ZnO system. For example, this is the case with ZnO hierarchical spheres composed of elementary nanoparticles deposited over RGO [160,161].

P-type response and room-temperature sensitivity were also reported for RGO functionalized with ZnO nanowires exposed to ammonia [162] and for RGO layers functionalized with hierarchical ZnO sheets composed by elementary nanoparticles exposed to NO_2_ [163]. In this latter case, the p-type conductivity of the RGO phase was suggested as the motivation of the p-type sensing mechanism exhibited by the composite, whose intensity is further enhanced by interface effects. A similar mechanism may be inferred for the first case owing to the large amount (about 50% wt) of RGO.

The transition from n- to p-type sensing was observed in NH_3_ sensing with an RGO-loaded SnO_2_ film as the RGO load was increased [164].

SnO_2_ nanowires have also been used with RGO for H_2_S sensing at room temperature. In this case, the functionalization process left Cu contamination, and this was proposed as an important feature for the observed selective response to H_2_S [165]. Tests carried out with other compounds, including NO_2_ and ethanol, indicated an n-type response, differently from the previously discussed ZnO-RGO system; however, in the SnO_2_-RGO case, the presence of Cu may also have a relevant role for these compounds, not only for H_2_S. Cu was intentionally added as dopant in the ZnO film of the RGO-ZnO composite for H_2_S sensing at room temperature [166].

Room-temperature NO_2_ sensing was also achieved with RGO-functionalized In_2_O_3_ nanorods [167]. In this case, the response was n-type and ascribed to a dominant role of the MOX nanowire, which is n-type, with the MOX-RGO interface having the role of promoting sensing capability at room temperature [167].

Nanowires have been further exploited in the form of hierarchical structures in conjunction with RGO. An example is given by hierarchical 3D mesocrystals composed by Cu_2_O nanowires deposited over RGO [168]. The sensing mechanism is based on p-p junctions, which featured p-type responses to NO_2_ at room temperature [168].

Numerical values of the measured responses *S* and the type of sensing mechanisms are resumed in Table 6.

**Figure 9 sensors-22-03351-f009:**
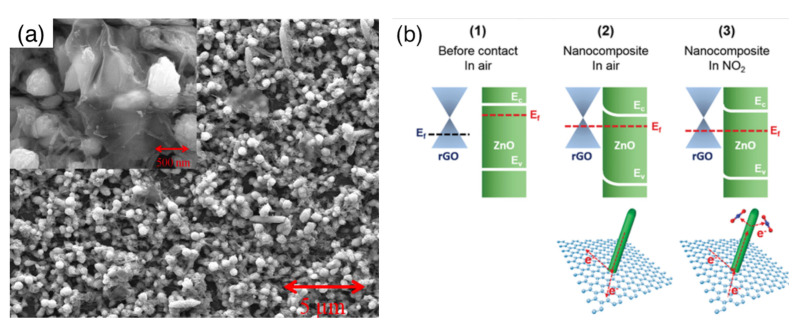
RGO–MOX interface. SEM image (top view) of the RGO–ZnO nanorods composite (**a**). Schematic representation of the electrical coupling mechanism between RGO and ZnO nanorods; energy bands are shown for the following three situations: disjointed materials, coupled materials exposed to air, and to NO_2_ (**b**). Figure 9a is reprinted from [169], Copyright (2018) with permission from Elsevier. Figure 9b is reprinted with permission from [159], Copyright 2016 American Chemical Society.

An additional mode to exploit the RGO–MOX interface has been proposed by Van Quang et al. They deposited SnO_2_ nanowires over both the electrodes of the device, leaving an empty gap between them (no nanowires between the electrodes). They further realized the connection between the two electrodes by means of a graphene sheet; in particular, the G sheet was not directly in contact with the metallic part of the electrodes but with the nanowires coating the electrodes. Owing to the metallic properties of graphene, the macroscopic electrical resistance of the device is given by the reversed biased junction formed at the G-SnO_2_ nanowire interface [170]. Similarly to the solution proposed by Wei et al. [139] discussed in Section 4.4.1, the sensing response is governed by the modulation of the Schottky barrier formed at the metal–semiconductor interface. In particular, the proposed G-SnO_2_ interface revealed the ability to detect NO_2_ at the ppb level. At the optimal temperature (150 °C), the device showed a response of about 6–50 ppb of NO_2_, which is comparable with the best performance reported in Table 1.

Before the advent of graphene and related materials, carbon was also widely exploited in the form of 1D carbon nanotubes. A recent article compared the SnO_2_-RGO and SnO_2_-CNT systems, in particular using semiconducting multiwalled CNTs. RGO functionalization revealed higher performance than that with CNTs, and this result was ascribed to the larger density of carbon–MOX interfaces for the 2D material. This enhancement was observed for SO_2_ detection, while both types of functionalization revealed similar improvements with respect to the pristine SnO_2_ using compounds such as CO and CH_4_ [171]. The motivation of this selectivity was not explained; it may reasonably arise from the complex dependence of both RGO and CNTs properties on their synthesis, defects, stresses and bending [172,173].

#### 4.5.2. Organic Receptors

In addition to carbon allotropes and their (partially)oxidized nanostructures, other organic materials have been exploited to functionalize the MOX support.

For example, ZnO functionalization was studied with tris(hydroxymethyl)aminomethane (THMA). The supporting material was a network of ZnO nanowires, over which a coating of dispersed ZnO nanoparticles was applied. Nanoparticles were fundamental to increase the surface area of the supporting ZnO layer available for the THMA molecules with respect to the case of ZnO nanowires [174]. Indeed, the responses to NO_2_ recorded with ZnO nanowires directly functionalized with THMA were quite similar to those recorded with the pristine oxide, while an increase of about two times was observed with the intermediate nanoparticle layer [174].

Recently, organosiloxanes have also been employed as organic coatings for acetone detection. A three- and a five-fold enhancement with respect to the pristine ZnO nanowire network were obtained using 3-glycidoxypro-pyltrimethoxysilane (GLYMO) and (3-aminopropyl)trimethoxysilane (APTMS), respectively [175]. As for GLYMO, the enhancement was ascribed to electronic and chemical sensitization. Similar to the mechanism described in Section 4.4, it arises from electrons transferring from the ZnO to the organic layer owing to Fermi levels’ alignment. Concerning APTMS, it also involves the receptor function of the device thanks to the ammine functional groups of APTMS. These groups remain available for interaction with gases and in particular with the carbon of acetone: C_3_H_6_O + (NH_2_)^−^ ⇔ (H_3_C)_2_–C=N + H_2_O, [175].

N-[3-(Trimethoxysilyl)propyl]ethylenediamine (en-APTAS 1) has been used to functionalize SnO_2_ nanowires for NO_2_ sensing, reaching a response of about 10 to 250 ppb [176]. From the calibration curve, a response of about 2 is extrapolated for the reference concentration of 50 ppb discussed in Section 4.1, which is comparable with the most performing NO_2_ chemiresistors reported in the literature and resumed in Table 1.

Another important class of organic materials for gas sensing is the family of porphyrins. These materials have been widely exploited in the field, with optical and mass-sensitive transducing mechanisms often preferred to the electrical one owing to the poor conductivity of these materials [177]. To exploit the receptor functionalities of these organic materials through an electrical transduction mechanism, porphyrin-functionalized MOX chemiresistors working at room temperature have been developed, using visible light illumination to activate the electrical coupling between the supporting MOX and the receptor. These devices have been developed based on ZnO nanoparticles [178] and ZnO nanorods [179], showing the capability of the organic receptor to tune the sensitivity of the material to selected chemicals. For example, while pure ZnO revealed similar responses to ethanol and triethylamine, its functionalization with the H2TPPCOOH-porphyrin revealed suitability for increasing the partial selectivity in favor of the latter compound [179]. Such a capability was further exploited to discriminate off odors released by beef meat during the spoilage process [180].

The results discussed in this Section are resumed in Table 7.

### 4.6. Self-Heating Effect

This working mode aims to exploit the current flowing through the sensitive layer to warm it by the Joule effect, without the use of an external heater. It was first tested in 2003 using a porous, polycrystalline film composed by μm-sized particles for CO detection [181]. At that time, besides the technological interest for the layout simplification achieved by avoiding the external heating element, the high density of grain boundaries intrinsic in the polycrystalline material was not effective in reducing power consumption. Indeed, the device required a power supply of the order of 1 W to activate CO sensing.

On the other hand, this idea revealed its full potential when it was tested with nanowire materials, in particular with chemiresistors based on a single-crystalline nanowire contacted by two electrodes. Indeed, as discussed in more detail in Section 2 and Section 3, the nanowire is an excellent conductive channel for charge carriers thanks to its single crystalline structure, free from current-limiting elements such as grain boundaries [1]. Considering the small mass of the nanowire, a small current flowing through the nanowire itself is enough to heat it till the typical working temperature of gas sensors. To achieve a suitable self-heating effect, the device structure should also be designed to reduce heat dissipation as much as possible. Indeed, efficient self heating is typically achieved working with suspended nanowires, i.e., nanowires that are not in direct contact with the substrate but are separated from it by a gap of air [54]. A schematic representation of the suspended nanowire exploiting the self-heating mode is shown in Figure 10a. With this configuration, a few tens of μW (with currents of a few tens of μA) allow to reach the desired temperature [182]. Simulations for the temperature distribution along the suspended and the not-suspended nanowire is shown in Figure 10b [183]. The power consumption is lowered by about four orders of magnitude with respect to ordinary, bulky sensor substrates, whose power consumption is typically of the order of a few 100 mW, and by about 3 decades with respect to MEMS-based chemiresistors, which require a few 10 mW [184,185]. Such impressive power saving enables the development of sensing systems that are able to exploit the thermally activated reactions described in Section 3.1, and at the same time are energetically autonomous. For example, a proof-of-concept device was realized by coupling the self-heated chemiresistor with a thermoelectric microgenerator [186].

Despite the most effective exploitation of the self-heating effect being achieved with a single-nanowire device, interfaces have also been introduced in a controlled way to take advantage of their functional properties. For example, depositing nanowires with controlled density between two closely separated electrodes, the self-heating effect was exploited with a networked layer. Figure 10c shows the schematic representation of this type of self-heated device. The reduced gap between electrodes (2 μm) and the wire morphology of the MOX nanostructures were important to ensure a network with a few crystallite–crystallite junctions, which are the current-limiting elements [187]. The reduced number of junctions allowed the exploitation of the self-heating effect to detect NO_2_ with about 10 μW. Increasing the density of the network results in an increase in the required power supply to the order of 1 mW due to the larger number of junction elements [187].

Self heating was also exploited with composite metal oxides to merge the low power consumption with benefits arising from the heterointerfaces of the composite. As an example, a low-density network composed of ZnO nanowires coated with a SnO_2_ shell and further functionalized by Pt nanoparticles was integrated in a device characterized by a short gap (a few μm) between the electrodes. The C_7_H_8_-sensing performance were optimized by tuning the ZnO coverage thickness, resulting in sub-ppm sensing capability with a power supply of about 30 μW [188].

Self-heated devices are also appealing owing to their fast thermal dynamics, which feature time constants of the order of ms [189], comparable with those exhibited by microhotplates [190]. This allows for the use of temperature-modulation techniques that have been widely investigated and exploited with both bulky and MEMS substrates to increase the selectivity of individual sensors [191,192]. Since the response of MOX strongly depends on the working temperature, the periodic modulation of this parameter allows the exploration of a range of different sensitivities and to emphasize the partial selectivity of MOX based on gas-interaction transients [193].

For example, a self-heated SnO_2_ single nanowire was used to track the CO concentration in a background with different humidity levels using a square-wave bias characterized by a duty cycle of 50% and period of 1 s [194].

The fast thermal time constants exhibited by self-heated devices have also been exploited to further reduce power consumption from the μW range down to the pW level. In particular, applying a short bias pulse (a few ms) every few seconds to a single SnO_2_ nanowire device, Meng et al. achieved sub-ppm sensing of NO_2_ using less than 40 pW [183].

## 5. Conclusions

The main approaches to develop resistive gas sensors exploiting metal oxide nanowires have been reviewed. About 20 years have passed since the first publications proposing the use of these materials for chemosensing as an alternative to polycrystalline layers composed by nanoparticles. The goal of this paper is to review the results achieved so far in the field, starting from the structural and morphological features that attracted the attention of the scientific community since the early works.

The overall impression is for the most effective exploitation of the nanowire structure and morphology being achieved in terms of power-supply saving and utility-factor optimization.

Regarding the former, the best results have been obtained through the self-heating effect, which exploits the absence of grain boundaries in the single-nanowire device and the Joule effect to reach the temperature required to activate the gas-sensing mechanism. This is probably the most impressive result, since it offers the opportunity to reduce the power consumption by 3–4 orders of magnitude with respect to state-of-the-art MEMS-based chemiresistors, hence opening enormous potentialities for integration in portable devices, including smartphones, and the development of networks of energy-autonomous sensors. Its prospects are mainly in two directions: (i) device integration, i.e., the development of methodologies and processes to integrate these sensors into effective sensing systems, often exploiting the electronic nose and/or the temperature-modulation approach; (ii) lowering the preparation cost. The exploitation of nanowire networks with a low and well-controlled number of nanowire–nanowire junctions is particularly promising in this sense, allowing for a suitable compromise between power saving and ease of device fabrication, which is still unsatisfactory with the actual preparation technology of single-nanowire devices.

The latter is an intrinsic feature in nanowire mats, which are typically characterized by wide-open volumes between wires, suitable for an efficient gas diffusion. Nonetheless, it should be noted that porosity optimization is intrinsic but not unique for nanowire mats. Indeed, several papers reported successful results achieved with nanoparticle layers, both organized in hierarchical structures and traditional thick films. Unique to nanowires is the branched structure, a particular form of hierarchical architecture, which allows for the merging of the open structure of nanowire networks with the functional properties of the MOX–MOX interfaces. Apart from this particular configuration, functionalization has not emerged as revealing something particular arising from the nanoparticle or nanowire morphology. Functionalization with metallic or metal oxide nanoparticles as well as organic materials has been applied with both types of MOX morphologies and may be used in combination with all the aforementioned configurations of nanowires to extend the possibilities to tune the sensing properties of the considered device. Especially concerning metallic and MOX catalysts, approaches traditionally developed with thick films as support are being further exploited and adapted for application with nanowire-based devices. Opportunities seem almost equally open for both nanowires and nanoparticles.

As for the different surface termination of nanowires and nanoparticles, flat vs. rounded, this was already highlighted in the first gas-sensing publications. Since then, it has been proposed as an appealing opportunity to merge the results of surface science, typically obtained by working with macroscopic single crystals, with those of applicative fields such as catalysis and gas sensing, which employ films composed by interconnected nanostructures. Despite the remarkable results obtained both in terms of sensor response and partial selectivity, a proper understanding of surface termination effects is still to be achieved and there is much room for both fundamental and device-oriented studies in order to reduce the gap between these two fields. The main results reviewed in the present paper are briefly summarized in Table 8.

## Figures and Tables

**Figure 1 sensors-22-03351-f001:**
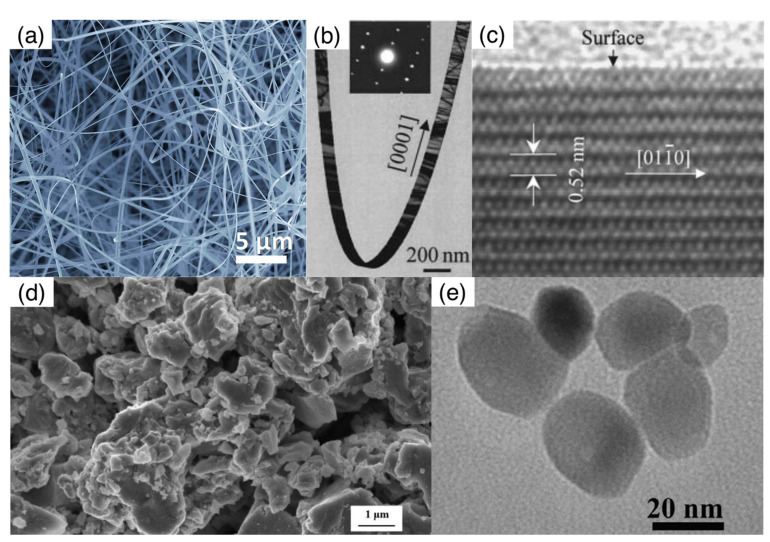
Comparison between the morphological features of nanowires and nanoparticles: (**a**) Scanning electron microscopy (SEM) picture of SnO_2_ nanowires/nanobelts; (**b**) transmission electron microscopy (TEM) image of a single ZnO nanobelt, the inset reports the selected area electron diffraction (SAED) pattern of the nanobelt; (**c**) high-resolution TEM (HR-TEM) image of the surface of a ZnO nanowire; (**d**) SEM image of a network of SnO_2_ nanoparticles; (**e**) TEM image of a few In_2_O_3_ nanoparticles. Figure 1a is reprinted from [12]. Figure 1b,c are from [1], reprinted with permission from AAAS. Figure 1d is reprinted from [13]. Figure 1e is reprinted from [14], Copyright (2016), with permission from Elsevier.

**Figure 3 sensors-22-03351-f003:**
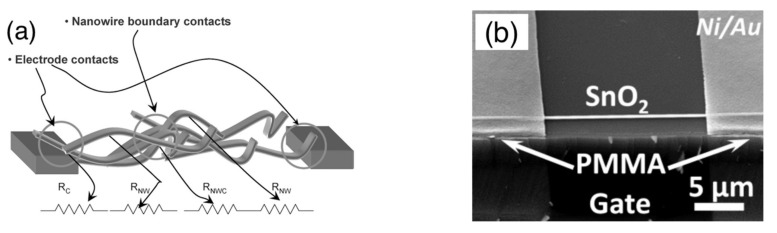
Types of nanowire-based chemiresistors: (**a**) Schematic representation of a chemiresistor based on a disordered network of nanowires and the main components of the equivalent electrical circuit; (**b**) SEM image of a chemiresistor based on a single nanowire contacted by two electrodes. Figure 3a is reprinted from [34], Copyright (2010), with permission from Elsevier. Figure 3b is from [12].

**Figure 4 sensors-22-03351-f004:**
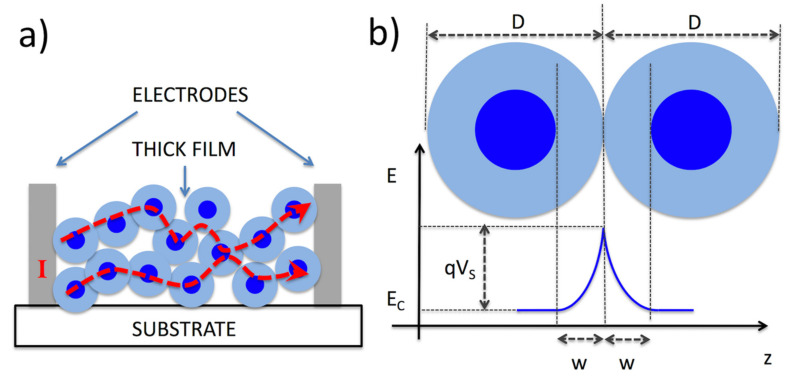
Schematic representation of the sensing mechanism of a thick-film-based gas sensor. The thick-film layout, composed by a network of interconnected particles (**a**) and the barrier E_b_ arising at the particle–particle junction (**b**). In each particle, the surface region depleted from charge carriers is distinguished by light-blue color from the inner portion (blue color) that maintains its unperturbed charge-carrier density. Figure 4a,b are adapted from [50].

**Figure 5 sensors-22-03351-f005:**
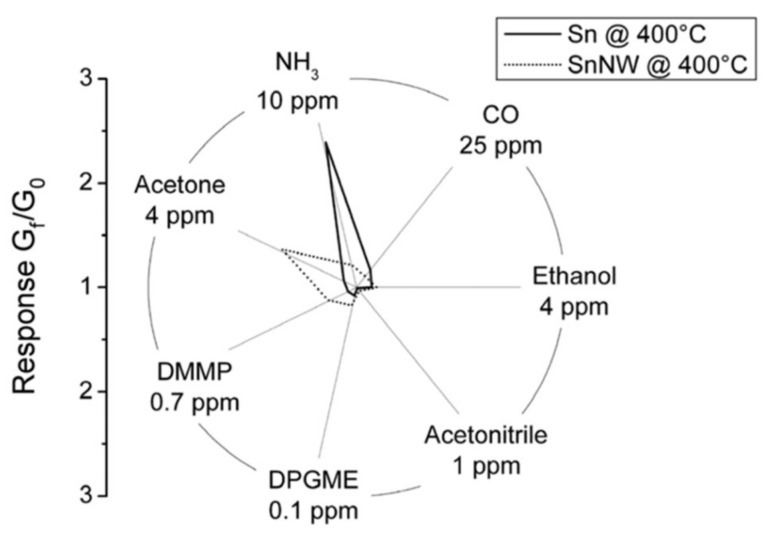
Surface termination effects in nanowire- and nanoparticle-based chemiresistors. Polar plot comparing the response of SnO_2_ nanowires (SnNW) and nanoparticles (Sn) against different chemicals. © 2008 IEEE. Reprinted, with permission, from [95].

**Figure 6 sensors-22-03351-f006:**
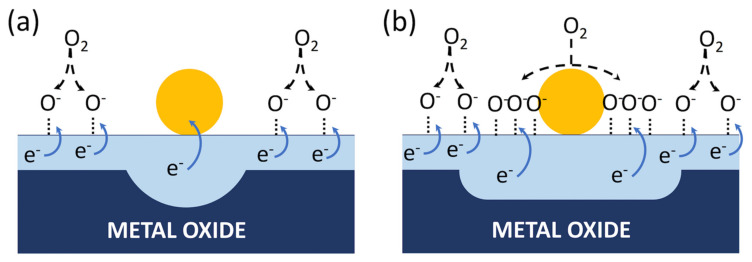
Schematic representation of metal oxide sensitization by means of metallic nanoparticles. (**a**) Electronic sensitization: the depletion layer is extended at the metal–metal oxide interface as a consequence of Fermi level alignment between the two materials; (**b**) spillover of oxygen molecules, O_2_, promoted by the metallic nanoparticle: oxygen molecules from the gas phase are dissociated by the metallic nanoparticle and further chemisorbed over the metal oxide surface. In the surrounding of the nanoparticle, this causes both an increase in the density of active ions and an extended depletion region.

**Figure 7 sensors-22-03351-f007:**
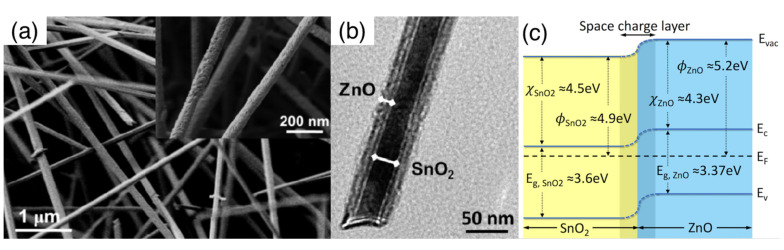
SEM (**a**) and TEM (**b**) images of SnO_2_-ZnO core–shell composite material and the schematic representation of the band structure for the SnO_2_-ZnO system (**c**). For both materials, the bulk values of their energy gap (E_g_), work function (*ϕ*), electron affinity (*χ*) are reported; profiles of the vacuum (E_vac_), valence band (E_v_), conduction band (E_c_) and Fermi level (E_F_) energies are also schematically reported. Figure 7a,b are reprinted from [148].

**Figure 10 sensors-22-03351-f010:**
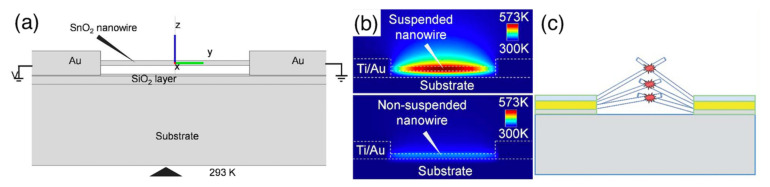
Self-heating. Schematic representation of the self-heating nanowire (**a**); simulation of temperature distribution along the nanowire for the suspended and not-suspended configurations (**b**); schematic representation of a self-heated nanowire network (**c**). Figure 10a,b are reprinted with permission from [183], Copyright 2016 American Chemical Society. Figure 10c is reproduced from Ref. [187] with permission from the Royal Society of Chemistry.

**Table 4 sensors-22-03351-t004:** Chemiresistors based on metal oxide (MOX) core–shell nanostructures. The response intensity S is calculated as S = R_gas_/R_air_ for NO_2_ and as S = R_air_/R_gas_ for other compounds in the case of n-type sensing response. The contrary is in the case of p-type sensing response. The n- or p- type response is reported in the S column. Gas concentrations are expressed in parts per million (ppm) and the sensor temperature is T.

Core MOX, Morphology	Shell MOX	T (°C)	Gas, Concentration	S (type)	Ref.
CuO, nanowire network	--	300	CO, 1 ppm	2.2 (p)	[145]
CuO, nanowire network	ZnO	300	CO, 1 ppm	6 (p)	[145]
ZnO, nanowire network	--	300	CO, 1 ppm	1.3 (n)	[145]
CuO, nanowire network	--	300	C_6_H_6_, 1 ppm	2.3 (p)	[145]
CuO, nanowire network	ZnO	300	C_6_H_6_, 1 ppm	5.8 (p)	[145]
ZnO, nanowire network	--	300	C_6_H_6_, 1 ppm	2.3 (n)	[145]
SnO_2_, nanowire network	--	300	C_7_H_8_, 10 ppm	2 (n)	[146]
SnO_2_, nanowire network	Cu_2_O	300	C_7_H_8_, 10 ppm	12 (p)	[146]
SnO_2_, nanowire network	--	300	C_6_H_6_, 10 ppm	2 (n)	[146]
SnO_2_, nanowire network	Cu_2_O	300	C_6_H_6_, 10 ppm	13 (p)	[146]
SnO_2_, nanowire network	--	300	NO_2_, 10 ppm	130 (n)	[146]
SnO_2_, nanowire network	Cu_2_O	300	NO_2_, 10 ppm	2 (p)	[146]
SnO_2_, nanowire network	--	300	CO, 10 ppm	5 (n)	[147]
SnO_2_, nanowire network	ZnO	300	CO, 10 ppm	80 (n)	[147]
SnO_2_, nanowire network	--	300	C_6_H_6_, 10 ppm	5 (n)	[147]
SnO_2_, nanowire network	ZnO	300	C_6_H_6_, 10 ppm	80 (n)	[147]
SnO_2_, nanowire network	--	300	NO_2_, 10 ppm	160 (n)	[147]
SnO_2_, nanowire network	ZnO	300	NO_2_, 10 ppm	25 (n)	[147]
ZnO, nanowire network	--	40	Triethylamine, 50 ppm	4 (n)	[149]
ZnO, nanowire network	SnO_2_	40	Triethylamine, 50 ppm	7 (n)	[149]
ZnO, nanowire network	SnO_2_ + Au	40	Triethylamine, 50 ppm	12 (n)	[149]
ZnO, nanowire network	--	40	Acetone, 500 ppm	2 (n)	[149]
ZnO, nanowire network	SnO_2_	40	Acetone, 500 ppm	5 (n)	[149]
ZnO, nanowire network	SnO_2_ + Au	40	Acetone, 500 ppm	6 (n)	[149]
ZnO, nanowire network	--	40	Ethanol, 50 ppm	2 (n)	[149]
ZnO, nanowire network	SnO_2_	40	Ethanol, 50 ppm	4 (n)	[149]
ZnO, nanowire network	SnO_2_ + Au	40	Ethanol, 50 ppm	6 (n)	[149]
Ga_2_O_3_, nanowire network	--	600	Ethanol, 1000 ppm	100 (n)	[150]
Ga_2_O_3_, nanowire network	SnO_2_	400	Ethanol, 1000 ppm	65 (n)	[150]

**Table 5 sensors-22-03351-t005:** Chemiresistors based on branched metal oxide (MOX) nanostructures. The response intensity S is calculated as S = R_gas_/R_air_ for NO_2_ and as S = R_air_/R_gas_ for other compounds in the case of n-type sensing response. The contrary is in the case of p-type sensing response. The n- or p- type response is reported in the S column. Gas concentrations are expressed in parts per million (ppm) and the sensor temperature is T.

Backbone MOX	Coating, Morphology	T (°C)	Gas, Concentration	S (type)	Ref.
SnO_2_	ZnO, shell	400	Ethanol, 20 ppm	20 (n)	[148]
SnO_2_	ZnO, shell + branch	400	Ethanol, 20 ppm	32 (n)	[148]
Cu_x_O	--	250	Acetone, 50 ppm	1.2 (p)	[151]
Cu_x_O	ZnO, shell	250	Acetone, 50 ppm	1.5 (n)	[151]
Cu_x_O	ZnO, shell + branch	250	Acetone, 50 ppm	6.5 (n)	[151]
SnO_2_	--	50	NO_2_, 1 ppm	--	[152]
SnO_2_	Bi_2_O_3_ branch	50	NO_2_, 1 ppm	3 (n)	[152]
SnO_2_	Bi_2_O_3_ branch + Pt nanoparticles	50	NO_2_, 1 ppm	28 (n)	[152]
SnO_2_	--	250	NO_2_, 1 ppm	10 (n)	[152]
SnO_2_	Bi_2_O_3_ branch	250	NO_2_, 1 ppm	50 (n)	[152]
SnO_2_	Bi_2_O_3_ branch + Pt nanoparticles	250	NO_2_, 1 ppm	--	[152]
SnO_2_	--	300	NO_2_, 20 ppm	2 (n)	[153]
SnO_2_	ZnO branch	300	NO_2_, 20 ppm	4 (n)	[153]
SnO_2_	ZnO branch + Au nanoparticles	300	NO_2_, 20 ppm	13 (n)	[153]
SnO_2_	--	300	NO_2_, 10 ppm	2 (n)	[154]
SnO_2_	ZnO branch	300	NO_2_, 10 ppm	5 (n)	[154]
SnO_2_	ZnO branch + Cr_2_O_3_ nanoparticles	300	NO_2_, 10 ppm	58 (n)	[154]
Sb-doped SnO_2_	SnO_2_, branched	300	Ethanol, 100 ppm	51 (n)	[155]

**Table 6 sensors-22-03351-t006:** Chemiresistors exploiting composite materials based on metal oxide (MOX) and 2D carbon nanostructures, namely graphene (G) and reduced graphene oxide (RGO). The response intensity S is calculated as S = R_gas_/R_air_ for NO_2_ and as S = R_air_/R_gas_ for other compounds in the case of n-type sensing response. The contrary is in the case of p-type sensing response. The n- or p- type response is reported in the S column. In this column, ‘sb’ stands for the response arising from the metal–semiconductor Schottky barrier modulation (Equation (11)). Gas concentrations are expressed in parts per million (ppm); the sensor temperature is T; ‘RT’ stands for ‘room temperature’.

MOX, Morphology	2D Carbon Material	T (°C)	Gas, Concentration	S (type)	Ref.
ZnO, nanorods	--	RT	NO_2_, 1 ppm	1.8 (n)	[159]
--	RGO	RT	NO_2_, 1 ppm	1.2 (p)	[159]
ZnO, nanorods	RGO	RT	NO_2_, 1 ppm	2.2 (p)	[159]
ZnO, nanosheets	--	RT	NO_2_, 50 ppm	6 (n)	[160]
ZnO, nanosheets	RGO	RT	NO_2_, 50 ppm	9 (n)	[160]
ZnO, hierarchical spheres	--	110	NO_2_, 1 ppm	4 (n)	[161]
ZnO, hierarchical spheres	RGO	110	NO_2_, 1 ppm	20 (n)	[161]
ZnO, hierarchical porous sheets	RGO	RT	NO_2_, 1 ppm	10 (p)	[163]
ZnO, nanoparticles	RGO	RT	NH_3_, 1 ppm	1.07 (p)	[162]
Cu doped SnO_2_, nanowires	RGO	RT	H_2_S, 50 ppm	33 (n)	[165]
Cu doped SnO_2_, nanowires	RGO	RT	NH_3_, 50 ppm	1.25 (n)	[165]
Cu doped SnO_2_, nanowires	RGO	RT	NO_2_, 50 ppm	1.5 (n)	[165]
Cu doped ZnO, nanorods	RGO	RT	H_2_S, 50 ppm	1.05 (p)	[166]
In_2_O_3_, nanorods	RGO	RT	NO_2_, 97 ppm	2.5 (n)	[167]
Cu_2_O, hierarchical mesocrystals	--	RT	NO_2_, 2 ppm	1.4 (p)	[168]
--	RGO	RT	NO_2_, 2 ppm	1.2 (p)	[168]
Cu_2_O, hierarchical mesocrystals	RGO	RT	NO_2_, 2 ppm	1.7 (p)	[168]
SnO_2_, nanowire	G	RT	NO_2_, 0.1 ppm	11 (sb)	[170]

**Table 7 sensors-22-03351-t007:** Chemiresistors based on metal oxide (MOX) nanostructures functionalized with organic molecules. The response intensity S is calculated as S = R_gas_/R_air_ for NO_2_ and as S = R_air_/R_gas_ for other compounds in the case of n-type sensing response. The contrary is in the case of p-type sensing response. The n- or p- type response is reported in the S column. Gas concentrations are expressed in parts per million (ppm) or parts per billion (ppb) and the sensor temperature is T. (*): under solar illumination, (+): under UV illumination; (§): under visible light illumination.

MOX, Morphology	Organic Coating	T (°C)	Gas, Concentration	S (type)	Ref.
ZnO, nanowires	--	190	NO_2_, 2 ppm	1.3 (n)	[174]
ZnO, nanowires	THMA	190	NO_2_, 2 ppm	1.2 (n)	[174]
ZnO, nanowires + nanoparticles	--	190	NO_2_, 2 ppm	1.22 (n)	[174]
ZnO, nanowires + nanoparticles	THMA	190	NO_2_, 2 ppm	1.44 (n)	[174]
ZnO, nanowires	--	300	Acetone, 50 ppm	30 (n)	[175]
ZnO, nanowires	GLYMO	300	Acetone, 50 ppm	90 (n)	[175]
ZnO, nanowires	APTMS	300	Acetone, 50 ppm	160 (n)	[175]
SnO_2_, nanowires	en-APTAS 1	RT (*)	NO_2_, 250 ppb	10 (n)	[176]
ZnO, nanoparticles	H2TPPCOOH porphyrin	RT	Pentanol, 60 ppm	1.1 (n)	[178]
ZnO, nanorods	--	RT (+)	Ethanol, 10^4^ ppm	1.01 (p)	[179]
ZnO, nanorods	H2TPPCOOH porphyrin	RT (§)	Ethanol, 10^4^ ppm	1.002 (n)	[179]
ZnO, nanorods	--	RT (+)	Triethylamine, 10^4^ ppm	1.01 (p)	[179]
ZnO, nanorods	H2TPPCOOH porphyrin	RT (§)	Triethylamine, 10^4^ ppm	1.8 (n)	[179]

**Table 8 sensors-22-03351-t008:** Resumed comparison between nanowires (NWs) and nanoparticles (NPs), relationship between the structural/morphological properties and the related gas-sensing phenomena and properties.

Single-Phase Materials		
Feature	Nanowires (NWs)	Nanoparticles (NPs)
Surface termination	Well-defined crystalline planes: Receptor/transducer functions (response intensity and partial selectivity) related to specific crystalline planes.	Rounded shape: Receptor/transducer functions (response intensity and partial selectivity) related to spherical/irregular surfaces.
Length-to-diameter aspect ratio	Large, often >10: Less effective than NPs in terms of surface-to-volume ratio and transducer function; NW Network: open morphology for optimal diffusion (enhanced response intensity and diffusion-related partial selectivity owing to enhanced utility factor); NW network: porosity does not decrease with decrease in the NW diameter; Individual NW: no grain boundaries for improved stability (no grain coarsening); Individual NW and low-density NW networks: self-heating effect for extremely low power consumption.	Almost unitary, ≈1: Most effective morphology for optimization of surface-to-volume ratio and transducer function (transition from surface-to-volume depletion regimes as the NP diameter decreases); Porosity may decrease with decrease in the NP diameter; Large density of grain boundaries (elements featuring the most effective transducer function and preferential gas adsorption).
Doping	Modulation of charge-carrier density (and Debye length); Modulation of transducer and receptor functions.	
	Eventual dopant-induced phase transition will turn the NW structure from single- to polycrystalline.	Increased thermal stability owing to hindered grain-coarsening phenomena; Possible dopant-induced phase transition.
Hierarchical structures	More open morphology for optimal diffusion (enhanced response intensity and diffusion-related partial selectivity owing to enhanced utility factor;	
**Heterostructure-based materials**		
**Surface functionalization**	**Nanowires (NWs)**	**Nanoparticles (NPs)**
metallic nanoparticles	Electronic and chemical sensitization for lowering optimal sensing temperature, increasing response intensity and partial selectivity.	
MOX nanostructures	Exploitation of the interface and the additive sensing capabilities for enhanced response intensity and partial selectivity.	
	Branched NW heterostructure merging the open hierarchical morphology with the interface and additive sensing capabilities.	--
Organic materials	Exploitation of the interface and the additive sensing capabilities to reduce the optimal working temperature, enhancing the response intensity and the partial selectivity.	

## Data Availability

Not applicable.

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
