# Peer review of "Metal Oxide Chemiresistors: A Structural and Functional Comparison between Nanowires and Nanoparticles"

_sensors, 2022, doi:10.3390/s22093351_

Round 1

Reviewer 2 Report

The author focused on two metal oxides used for years in sensor technology, i.e. zinc oxide ZnO and tin oxide SnO2 and completely omitted the important compound which is gallium oxide Ga2O3. There is also tungsten oxide used for years, as well as copper oxide, especially in combinations with tin oxide or zinc oxide, as well as oxide heterostructures, including core-shell.

It is difficult to talk about the novelty of research or analysis techniques, because it is a review article discussing a number of literature reports (167 items) on chemical sensors made on the basis of zinc oxide, tin oxide and tungsten and copper oxides over the last twenty years. These types of publications are also necessary and useful, especially for someone who starts working with various types of metal oxides as a gas-sensitive material.

Remarks:

  • The article needs careful review to eliminate typos in the text (and in the captions of the drawings).
  • Some of the captions of the drawings contain full bibliographic descriptions of the source, which makes it difficult to read.
  • In the References there are incomplete bibliographic descriptions of the sources.

Reviewer 3 Report

This is a comprehensive review which is well organized. Therefore, it will contribute interesting informations related to comparision between metal oxide nanowires and nanoparticles in development of chemiresistors. However, this manuscript has to be upgraded following some minor issues

  1. A summaried table about properties, advantages, disadvantages, etc of metal oxide nanowires and nanoparticles in design of chemiresistors should be provided
  2. Table 1, S is sensitivity, it should be added unit like %/ppm. Limit of detection also need to insert
  3. Section 4.5. Inorganic-organic heterostructures is not apropriate in this manuscript. In this section, the author provide a composite structure. I think it is out of the topic and title of this review
